# Studying the 3d Ising surface CFTs on the fuzzy sphere

Zheng Zhou (周正)[1,2] and Yijian Zou[1]

**1** Perimeter Institute for Theoretical Physics, Waterloo, Ontario, Canada N2L 2Y5
**2** Department of Physics and Astronomy, University of Waterloo, Waterloo, Ontario, Canada N2L 3G1

4 August, 2024

## Abstract

**Boundaries not only are fundamental elements in nearly all realistic physical systems, but also greatly enrich the structure of quantum field theories. In this paper, we demonstrate that conformal field theory (CFT) with a boundary, known as surface CFT in three dimensions, can be studied with the setup of fuzzy sphere. We consider the example of surface criticality of the 3d Ising CFT. We propose two schemes by cutting a boundary in the orbital space and the real space respectively to realise the ordinary and the normal surface CFTs on the fuzzy sphere. We obtain the operator spectra through state-operator correspondence. We observe integer spacing of the conformal multiplets, and thus provide direct evidence of conformal symmetry. We identify the ordinary surface primary $o$, the displacement operator $D$ and their conformal descendants and extract their scaling dimensions. We also study the one-point and two-point correlation functions and extract the bulk-to-surface OPE coefficients, some of which are reported for the first time. In addition, using the overlap of the bulk CFT state and the polarised state, we calculate the boundary central charges of the 3d Ising surface CFTs non-perturbatively. Other conformal data obtained in this way also agrees with prior methods.**

# 1   Introduction

Boundaries are co-dimension one surfaces that naturally exist in realistic systems. They have played an important role in quantum field theory and condensed matter systems. The interplay of the bulk and boundary physics has led to great insight into topological phases [1], anomalies [2], and conformal field theories [3–6]. For example, in symmetry-protected topological phases, the boundary either spontaneously breaks the symmetry or becomes gapless due to the anomalous action of the symmetry [7]. In conformal field theory, the classification of conformal boundary condition is much more intricate and is only solved in limited cases such as $(1+1)$d rational CFTs and free theories [8–10]. In higher dimensions, much less is known non-perturbatively.

We consider the CFT on the semi-infinite Euclidean spacetime with $x_\perp > 0$. Due to the existence of the boundary at $x_\perp = 0$, the symmetry group is broken from the conformal group in $2+1$ dimensions, SO(4, 1) to the global conformal group SO(3, 1) in $1+1$ dimensions. As a consequence, there exist a set of operators living on the surface that transforms under the representation of the remaining conformal symmetry SO(3, 1) [11]. In particular, these operators have scaling dimensions $\hat{\Delta}$, transverse spin $l_z$ under SO(2) rotation, and possess the structure of primaries and descendants that resembles the bulk CFT. Some correlation functions forbidden by the bulk conformal symmetry can also be non-zero in the surface CFT due to the reduced conformal symmetry [4]. Thus, a surface CFT is characterised by a richer set of conformal data. The simplest example is the one-point function of a bulk scalar primary

$$\langle \phi(x) \rangle = \frac{a_\phi}{|2x_\perp|^{\Delta_\phi}} \tag{1}$$

where $x_\perp$ is the perpendicular distance to the surface, $\Delta_\phi$ is the scaling dimension of $\phi$, and $a_\phi$ is the bulk one-point OPE coefficient, a universal data in the surface CFT. Moreover, the form of the two-point function between a bulk scalar primary $\phi$ and a surface scalar primary $\hat\phi$ is fixed by the surface conformal symmetry

$$\langle \phi(x)\hat\phi(0)\rangle = \frac{b_{\phi\hat\phi}}{|2x_\perp|^{\Delta_\phi - \Delta_{\hat\phi}}|x|^{2\Delta_{\hat\phi}}} \tag{2}$$

where the bulk-to-surface two-point OPE coefficient $b_{\phi\hat\phi}$ is also universal in the surface CFT.

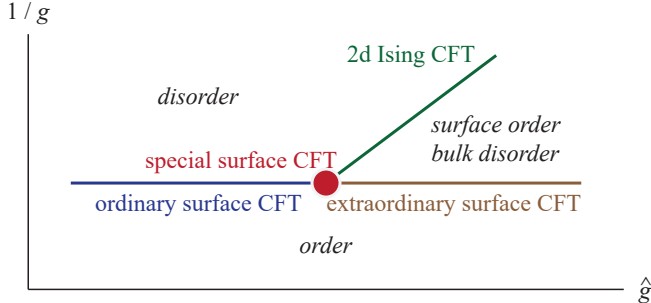

Figure 1: The phase diagram of the 3d Ising model with boundary in the parameter space of a bulk coupling constant $g$ and a surface coupling constant $\hat g$.

Here we specifically focus on the boundary of 3D Ising CFT which describes the universality class of $\mathbb{Z}_2$ symmetry breaking phase transition in 3D [12–16]. First, acting a pinning field that explicitly breaks the global $\mathbb{Z}_2$-symmetry on the boundary of the Ising CFT action $S = S_{\mathrm{CFT}} + \int_{x_\perp=0} \mathrm{d}^2x\, \sigma(x)$, one reaches the 'normal surface CFT' [17]. One can then consider the phase diagram when the $\mathbb{Z}_2$ global symmetry is not explicitly broken. As a minimal example, we consider a 3d lattice model with Ising spins $\sigma_i$ on half-infinite cubic lattice

$$\beta H = -g \sum_{\langle ij\rangle_{\mathrm{bulk}}} \sigma_i\sigma_j - \hat g \sum_{\langle ij\rangle_{\mathrm{boundary}}} \sigma_i\sigma_j. \tag{3}$$

The phase diagram when tuning the bulk coupling constant $g$ and a coupling $\hat g$ on the surface that respects the global $\mathbb{Z}_2$-symmetry is sketched in Figure 1. When the bulk is tuned to the critical point $g = g_c$, at small surface coupling constant $\hat g < \hat g_c$, the $\mathbb{Z}_2$-symmetry is preserved on the surface. This surface criticality corresponds to the free boundary condition on the surface and is described by the 'ordinary surface CFT'. When $g = g_c$ and $\hat g > \hat g_c$, the $\mathbb{Z}_2$-symmetry is spontaneously broken on the surface. This surface CFT is called the 'extraordinary surface CFT' [18, 19]. It is the direct sum of two copies of normal surface CFT with the surface spins polarised to opposite $\mathbb{Z}_2$ directions and can be therefore described by the same conformal data as the normal surface CFT. The transition point between ordinary and extraordinary CFT $g = g_c, \hat g = \hat g_c$ is described by the 'special surface CFT'. Unlike the ordinary and the extraordinary surface CFTs that are stable against surface perturbations, the special surface CFT has one relevant surface perturbation towards ordinary or extraordinary surface CFTs. If we fix $\hat g > \hat g_c$ and decrease the bulk coupling constant $g$ from the extraordinary surface CFT, the system will go through a phase $g_{c,s} < g < g_c$ where the bulk is disordered and the surface is ordered, and then a surface order-to-disorder transition at $g_c = g_{c,s}$ that is described by the 2d Ising CFT.

The three surface CFTs have been extensively studied by analytical and numerical methods such as perturbative calculation [20–23], conformal bootstrap [24, 25] and Monte Carlo [26–28]. The two-loop perturbative calculation [23] has obtained the the scaling dimensions of the lowest $\mathbb{Z}_2$-odd operator $\Delta_o = 1.26$ in the ordinary surface CFT. Large-$N$ calculation [13, 14, 29] has also reported boundary central charge of ordinary $c_{\text{bd}}^{(O)} = -1/16 + \mathcal{O}(N^{-1})$ and normal $c_{\text{bd}}^{(N)} = -9/16 + \mathcal{O}(N^{-1})$ surface CFTs. Both the ordinary and the normal surface CFT have been studied using the conformal bootstrap [24]. For the ordinary surface CFT, they have obtained $\Delta_o = 1.276(2)$ and the one-point bulk-to-surface OPE $a_\epsilon = 0.750(3)$ and $a_{\epsilon'} = 0.79(2)$. For normal surface CFT they obtain several one-point and two-point bulk-to-surface OPEs $a_\epsilon = 6.607(7), a_\sigma = 2.599(1), b_{\epsilon \text{D}} = 1.742(6), b_{\sigma \text{D}} = 0.25064(6)$[1]. On the other hand, by constructing corresponding lattice models, the Monte Carlo calculations have obtained the scaling dimensions of the lowest $\mathbb{Z}_2$-odd operator in the ordinary surface CFT $\Delta_o = 1.2751(6)$ [27] and the lowest $\mathbb{Z}_2$-even and $\mathbb{Z}_2$-odd operators in the special surface CFT $\Delta_{s,-} = 0.3535(6)$ and $\Delta_{s,+} = 1.282(2)$ [28]. The Monte-Carlo simulations also suffer from some disadvantages. For example, the conformal symmetry does not manifest, and it is also hard to obtain higher primaries.

Recently, fuzzy sphere regularisation has emerged as a new powerful method to tackle critical phenomena in three-dimensions [30–39]. By studying quantum systems on fuzzy (noncommutative) sphere [40] with geometry $S^2 \times \mathbb{R}$, the method realises $(2 + 1)$d quantum phase transitions. Compared with conventional methods which involve simulating lattice models, this approach offers distinct advantages including exact preservation of rotation symmetry, direct observation of emergent conformal symmetry and the efficient extraction of conformal data [41, 42].

In the fuzzy sphere method, the state-operator correspondence plays an essential role. Specifically, there is a one-to-one correspondence between the eigenstates of the critical Hamiltonian on the sphere and the CFT operators, where the energy gaps are proportional to the scaling dimensions. The power of this approach has been demonstrated in the context of the 3D Ising transition, where the presence of emergent conformal symmetry has been convincingly established [30]. It has also been extended to obtain the spectrum of defect operators for the 3d Ising model with the magnetic line defect [33] as well as the defect changing operators and $g$-functions [33, 35].

In this paper, we study the surface critical phenomena in the 3d Ising CFT with fuzzy sphere. One scheme (hereafter referred to as *'real space boundary scheme'*) to realise the surface CFTs is to add a pinning field in the real space to remove the degrees of freedom on the southern hemisphere. We further propose a simpler scheme (hereafter referred to as *'orbital space boundary scheme'*) to realise the surface CFTs by pinning the electrons on the $m < 0$ orbitals based on the observation that the LLL orbitals are localised around certain latitude circles in real space. We realise the ordinary and the normal surface CFTs in both schemes. We study the operator spectra of the two surface CFTs by diagonalizing the low-energy eigenstates of the fuzzy-sphere Hamiltonian, which correspond to scaling operators due to the state-operator correspondence. We have identified the surface primary $o$ in the ordinary surface CFT, the displacement operator D in both surface CFTs and their descendants up to $\Delta \leq 7, l_z \leq 4$. Their scaling dimensions agree perfectly with the prediction of conformal symmetry and also match the results of former studies using conformal bootstrap and Monte Carlo after we do a careful finite-size analysis. We also study the one-point

---

[1]In this paper we adopt a different convension for normalising D (*cf.* Section 4).

and two-point correlation functions. The correlation functions calculated with the fuzzy sphere approach to the prediction of conformal symmetry by increasing system sizes. We then calculate the bulk one-point and bulk-to-surface two-point OPE coefficients. Our results match the results by conformal bootstrap and also agree with the prediction of the Ward identity which constraints OPE coefficients involving displacement operators. Some of the OPE coefficients like $b_{\epsilon D}$ in the ordinary surface CFT have not been previously calculated and are reported for the first time. An alternative approach to calculate the conformal data of surface CFT is the wavefunction overlap method [35, 43–45]. By computing the overlap between a bulk CFT state and a polarised state, we obtain several bulk-to-surface OPE coefficients and also the boundary central charge, where the latter has not been computed non-perturbatively in the literature. Our main results are listed in Table 1.

Table 1: A table of the main results in this paper for the ordinary and normal surface criticalities in the 3d Ising CFT, including the scaling dimension of the ordinary surface primary $o$, various one-point and two-point OPE coefficients, the Zamolodchikov norm $C_D$ and the boundary central charge $c_{bd}$. The error bars are estimated from finite-size extrapolation. Throughout this paper, we take the error bar from finite-size scaling to be the difference between the extrapolated value and the data of the largest possible system size. We also need to note that the error bars given are not strict but only an estimation.

| Ordinary | | Normal | |
|---|---|---|---|
| $\Delta_o$ | 1.23(4) | $a_\epsilon$ | 6.4(9) |
| $a_\epsilon$ | 0.74(4) | $a_\sigma$ | 2.58(16) |
| $b_{\epsilon D}$ | 0.92(4) | $b_{\epsilon D}$ | 1.74(22) |
| $b_{\sigma o}$ | 0.87(2) | $b_{\sigma D}$ | 0.254(17) |
| $C_D$ | 0.0089(2) | $C_D$ | 0.176(2) |
| $c_{bd}$ | −0.0159(5) | $c_{bd}$ | −1.44(6) |

This paper will be organised as follows:

• In Section 2, we review the setup of the fuzzy sphere and propose the two schemes to realise the ordinary and normal surface CFTs on the fuzzy sphere based on cut in real space and orbital space.

• In Sections 3 and 4, we present the results in the orbital space boundary scheme. In Section 3, we present the operator spectrum of the ordinary and normal surface CFT.

• In Section 4, we present the one-point and two-point correlation functions and the OPE coefficients.

• In Section 5, we present the results in the real space boundary scheme and find agreement with the orbital-space scheme.

• In Section 6, we introduce an alternative approach to calculate the boundary central charge and other conformal data of surface CFT using the overlap between a bulk CFT state and a polarised state.

• In Section 7, we make a brief discussion and summarise.

## 2  Model

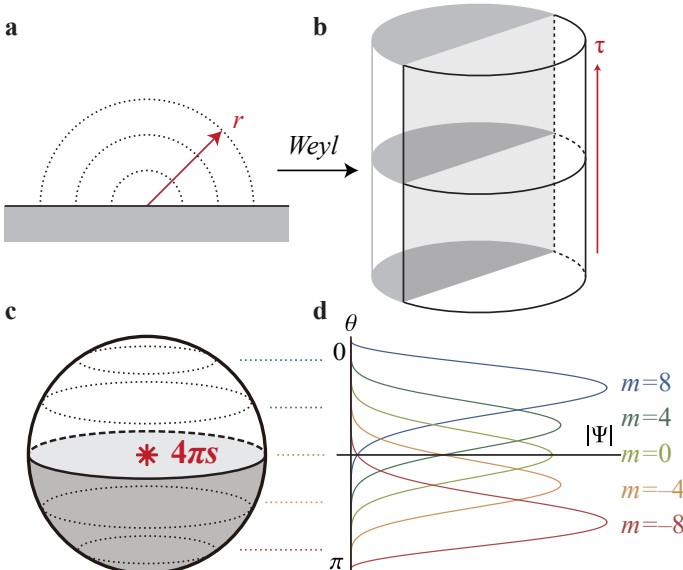

Figure 2: (a) An illustration of the path integral of surface CFTs on the semi-infinite flat spacetime and (b) its Weyl transformation onto cylinder. (c) An illustration of the fuzzy sphere setup to realise surface CFTs. (d) The spatial probability density distribution $|Y_{sm}^{(s)}(\theta, \phi)|^2$ of the LLL orbitals for $s = 10$

.

We start by introducing the realisation of the bulk 3d Ising CFT on the fuzzy sphere [30]. The setup involves electrons with isospin-$1/2$ moving on a sphere in the presence of $4\pi s$-monopole at its centre (Figure 2c) [40]. Due to the presence of the monopole, the single-particle eigenstates form highly degenerate quantised Landau levels. The ground state, *i.e.*, lowest Landau Level (LLL) has a degeneracy $N = 2s+1$ for each isospin. We half-fill the LLL and set the gap between the LLL and higher Landau levels to be much larger than other energy scales in the system. In this case, we can effectively project the system into the LLL. After the projection, the coordinates of electrons are not commuting anymore. We thus end up with a system defined on a fuzzy (non-commutative) two-sphere, and the linear size of the sphere $R$ is proportional to $\sqrt{N}$. The 3d Ising transition can be realised by adding interactions that mimic a $(2 + 1)$d transverse field Ising model on the sphere

$$H_{\text{bulk}} = \int d\Omega_1 d\Omega_2\, U(\theta_{12})[n^0(\Omega_1)n^0(\Omega_2) - n^z(\Omega_1)n^z(\Omega_2)] - h \int d^2\Omega\, n^x(\Omega). \qquad (4)$$

The density operators are defined as $n^\alpha(\Omega) = \Psi^\dagger(\Omega)\sigma^\alpha\Psi(\Omega)$, where $\Psi(\Omega) = (\psi_\uparrow(\Omega), \psi_\downarrow(\Omega))^{\text{T}}$, $\sigma^{x,y,z}$ are the Pauli-matrices and $\sigma^0 = \mathbb{I}_2$. The density-density interaction potential $U(\Omega_{12})$ and the transverse field $h$ are taken from Ref. [30] to ensure the bulk realises the Ising CFT. In practice, the second quantised operators can be expressed in terms of the annihilation operator in the orbital space as $\Psi(\Omega) = N^{-1/2} \sum_{m=-s}^{s} Y_{sm}^{(s)}(\hat{r})\mathbf{c}_m$, where the monopole spherical harmonics

$$Y_{sm}^{(s)}(\hat{r}) = \frac{e^{im\phi} \cos^{s+m}\frac{\theta}{2} \sin^{s-m}\frac{\theta}{2}}{\sqrt{4\pi \mathrm{B}(s + m + 1, s - m + 1)}} \qquad (5)$$

is the single particle wavefunction of the LLL, and B is the Euler beta function. The Hamiltonian can be expressed as quadratic and quartic terms of the creation and annihilation operators $\mathbf{c}_m^{(\dagger)}$ as well.

Taking into account the imaginary time direction as well, the path integral configuration of the quantum system on the sphere lives on a geometry of cylinder $S^2 \times \mathbb{R}$. This is conformally equivalent to the flat spacetime $\mathbb{R}^3$ through a Weyl transformation $(\hat{\mathbf{n}}, \tau) \in S^2 \times \mathbb{R} \mapsto e^{\tau} \hat{\mathbf{n}} \in \mathbb{R}^3$ where $\hat{\mathbf{n}}$ is the unit vector on the 2-sphere, which maps the time slice of the cylinder to concentric spheres in the flat spacetime. Conversely, a surface CFT that lives on a half infinite spacetime $z > 0$ is equivalent to a quantum system on a hemisphere through radial quantisation (Figure 2a,b).

Hence, to realise the surface criticalities in 3d Ising CFT, we need to remove the degrees of freedom on the southern hemisphere $z = \cos \theta < 0$. The most straightforward way is to add a homogeneous pinning field

$$H_{\text{surface, space}} = \int_{z<0} d\Omega \, \mathbf{h}_s \cdot \mathbf{n}(\Omega) = \sum_m B_{1/2}(s - m + 1, s + m + 1) \mathbf{h}_s \cdot c_m^{\dagger} \sigma c_m. \tag{6}$$

where $B_x(\alpha, \beta)$ is the incomplete beta function[2]. Different states that the southern hemisphere is pinned to correspond to different boundary conditions. To realise the two surface CFTs, we make two choices of $\mathbf{h}_s$:

1. When $\mathbf{h}_s$ points along the $+x$-direction, this corresponds to a free boundary condition and will result in the ordinary surface CFT;

2. When $\mathbf{h}_s$ points along the $z$-direction, this explicitly breaks $\mathbb{Z}_2$-symmetry and results in a boundary order, which will result in the normal surface CFT in the 3d Ising criticality.

In the following, we will refer to this scheme as real space boundary. It is a little bit technical to realise this scheme on the fuzzy sphere, as one needs to transform the operators in the real space to the orbital space.

We propose a simpler realisation by noticing that the LLL orbitals are localised in the real space at bands in the vicinity of latitude circles (Figure 2d). The average latitude and angular width of each orbital

$$\cos \bar{\theta}_m = \langle \cos \theta \rangle_m = \int d\Omega \, |Y_{sm}^{(s)}|^2 \cos \theta = \frac{m}{s+1}$$

$$\delta\theta_m = \frac{\sqrt{\langle \cos^2 \theta \rangle_m - \langle \cos \theta \rangle_m^2}}{\sin \bar{\theta}_m} = \frac{1}{\sqrt{2s+3}} \tag{8}$$

shows the following two conclusions:

1. the angular width of the band scales to zero as $R^{-1} \sim N^{-1/2}$ in the thermodynamic limit; and

2. the orbitals are distributed evenly from the north pole to the south pole in the same order as $m$; in particular, the $m = \pm s$ orbitals are distributed in the vicinity of the north and south pole and the $m = 0$ orbital distributes in the vicinity of the equator.

---

[2]To derive this expression, we can perform the integral explicitly

$$\int_{z<0} d\Omega \, \mathbf{h}_s \cdot \mathbf{n}(\Omega) = \sum_{mm'} \mathbf{h}_s \cdot c_{m'}^{\dagger} \sigma c_m \int_{z<0} d\Omega \, \bar{Y}_{sm'}^{(s)} Y_{sm}^{(s)} = \sum_{mm'} B_{1/2}(s - m + 1, s + m + 1) \delta_{mm'} \mathbf{h}_s \cdot c_{m'}^{\dagger} \sigma c_m. \tag{7}$$

Hence, most of the $m > 0$ orbitals are distributed only in the northern hemisphere, and most of the $m < 0$ orbitals are distributed only in the southern hemisphere, except a small portion of orbitals that passes through the equator whose number scales subextensively with $R \sim N^{1/2}$, *i.e.*, with the length of the boundary. Thus, we can make a boundary cut in the orbital space instead of the real space and freeze the degrees of freedom at $m < 0$. In other words, we can add an infinite pinning field

$$H_{\text{surface, orbital}} = \lim_{h_s \to \infty} \sum_{m<0} h_{s,\alpha} \mathbf{c}_m^\dagger \sigma^\alpha \mathbf{c}_m, \tag{9}$$

where $\alpha = 0, x, y, z$. In the thermodynamic limit, this is equivalent to making a boundary cut in the real space. In particular, the possible choices of the pinned orbitals are

1. setting the $m < 0$ orbitals to be all empty, which corresponds to free boundary condition and the ordinary surface CFT;

2. setting the $m < 0$ orbitals to be half-filled and polarising them to the $+x$-direction, which also corresponds to the ordinary surface CFT; and

3. setting the $m < 0$ orbitals to be half-filled and polarising them towards $z$-direction, which corresponds to the normal surface CFT.

In practice, we only have to keep the $m > 0$ degrees of freedom, because the effect of pinned $m < 0$ orbitals can be written equivalently as a polarising field acting on the remaining $m > 0$ orbitals (*cf.* Appendix A). In the following, we will refer to this scheme as orbital space boundary.

Even though the existence of the boundary explicitly breaks the sphere rotation symmetry, there is a large group of remaining symmetries which we list below.

1. The global $\mathbb{Z}_2$-symmetry is kept in the realisations of the ordinary surface CFT. Each operator will thus carry a $\mathbb{Z}_2$ quantum number. This symmetry is broken in the realisations of the normal surface CFT.

2. The SO(3) rotation symmetry is broken to U(1) rotation symmetry with respect to the $z$-axis. Each operator will thus carry a quantum number $l_z$. The operators $\partial = \partial_x + i\partial_y$ and $\bar{\partial} = \partial_x - i\partial_y$ send an operator to its descendants with $l_z$ increased or decreased by 1.

3. The particle-hole symmetry $\mathscr{P} : \mathbf{c}_m \mapsto i\sigma_y(\mathbf{c}_m^\dagger)^{\text{T}}$ that corresponds to the parity in the CFT is preserved. It now becomes the improper $\mathbb{Z}_2$ of the O(2) rotation symmetry and connects an operator with $+l_z$ to an operator with $-l_z$, ensuring that the spectrum is symmetric when reflected with respect to $l_z$ axis[3].

We solve the low-lying spectrum of the system numerically through exact diagonalisation (ED) or DMRG via the ITensor package [46, 47]. For the orbital-space boundary scheme, ED calculations are performed up to $N = 30$ and DMRG is performed up to system size $N = 64$ (*i.e.*, simulate $N/2 = 32$ orbitals in DMRG) with maximal bond dimension $D = 6000$. For the real-space boundary scheme, ED is performed up to $N = 16$ and DMRG is performed up to $N = 32$ (*i.e.*, simulate 32 orbitals in DMRG). For the DMRG calculation, the maximal truncation error for the largest size and bond dimension is $4 \times 10^{-8}$ (*cf.* Appendix C.1).

---

[3]A subtlety is that for the orbital space boundary condition that all the $m < 0$ orbitals are set to be empty, the particle-hole symmetry acts only on the $m > 0$ orbitals and send $l_z$ to $N^2 - l_z$. We then shift the quantum number by $N^2/2$ to guarantee the mirror symmetry of the spectrum.

# 3 Orbital space boundary scheme: Operator spectra

We first study the operator spectra of the surface CFTs in the orbital space boundary scheme. The operator spectra can be easily obtained in the fuzzy sphere through the state-operator correspondence [42], *i.e.*, the eigenstates of the quantum Hamiltonian on the hemisphere are in one-to-one correspondence with the scaling operators of the boundary operator of the surface CFT. The states and the corresponding operators have the same quantum numbers. The scaling dimensions and energies are related by

$$\Delta_{\hat{\phi}} = \frac{v}{R}(E_{\hat{\phi}} - E_{\hat{0}}), \tag{10}$$

where we use $\hat{\cdot}$ to denote the surface operators and states. This applies to both surface primaries and surface descendants in the form of $\partial^m \bar{\partial}^{\bar{m}} \hat{\phi}$ with scaling dimension $\Delta_{\hat{\phi}} + m + \bar{m}$ and transverse spin $l_z = m - \bar{m}$. The speed of light $v$ depends on the UV details of the model. Its value is identical in the bulk CFT, which can be determined by calibrating the conserved stress tensor with scaling dimension exactly 3: $v/R = 3/(E_{T^{\mu\nu}} - E_0)$. In finite-size systems, the calculated scaling dimensions are subject to finite-size corrections induced by irrelevant surface operators $\hat{S}$ in the symmetry singlet sector

$$\Delta_{\hat{\phi}}(N) = \Delta_{\hat{\phi}} + \sum_{\hat{S}} \lambda_{\hat{S};\hat{\phi}} R^{2-\Delta_{\hat{S}}}. \tag{11}$$

For the Ising surface CFTs, the lowest correction comes from the displacement operator D with scaling dimension $\Delta_D = 3$ and its descendant $\partial\bar{\partial}D$ with scaling dimension 5 [4]. Thus,

$$\Delta_{\hat{\phi}}(N) = \Delta_{\hat{\phi}} + \lambda_{D;\hat{\phi}} N^{-1/2} + \lambda_{\partial\bar{\partial}D;\hat{\phi}} N^{-3/2} + \mathcal{O}(N^{-3/2}) \tag{12}$$

To eliminate the contributions from those irrelevant operators, one way is to perform finite size scaling on data with different system sizes. Another more sophisticated way, following the recent work [49] by Lao and Rychkov, is to make use of conformal perturbation theory on the whole spectrum, utilising the fact that the corrections from an irrelevant singlet on a surface primary and its descendants are not independent. The latter way allows us to calculate the coefficients $\lambda_{\hat{S};\hat{\phi}}$ numerically, reducing finite-size corrections significantly as first explored by Ref [49] (*cf.* Appendix B).

## 3.1 Spectrum of ordinary surface CFT

We first study the spectrum of ordinary surface CFT. Perturbative calculation [23], lattice simulation [26, 27] and conformal bootstrap [24] has reported the lowest $\mathbb{Z}_2$-odd primary operator at $\Delta_o = 1.2751(6)$, and the lowest $\mathbb{Z}_2$-even primary operator being the displacement operator at $\Delta_D = 3$.

On fuzzy sphere, we rescale the excitation energies of the lowest lying states and do a finite size scaling according to Eq. (12) up to the $N^{-1/2}$ order both in the boundary condition that $m < 0$ orbitals are empty and that they are half-filled and polarised towards $+x$-direction. The results are

---

[4]We note that the bulk $\mathbb{Z}_2$-even irrelevant operators also contribute to the finite-size effect. However, the lowest irrelevant operators $\epsilon'$ is tuned away, and the second lowest one $C_{l=4}$ has scaling dimension 5.02 [48], so its contribution decays faster than D.

plotted in Figure 3 and listed in Table 3 in the Appendix C.2. We find that the scaling dimensions in both boundary conditions scale to the same value in the thermodynamic limit. We estimate that the lowest $\mathbb{Z}_2$-odd primary has scaling dimension

$$\Delta_o = 1.23(4),$$

where the error bar is estimated from finite-size extrapolation. Throughout this paper, we take the error bar from finite-size scaling to be the difference between the extrapolated value and the data of the largest possible system size. We also need to note that the error bars given are not strict but only an estimate. The value of $\Delta_o$ is consistent with the estimate from conformal bootstrap $\Delta_{o,\text{CB}} = 1.276(2)$ and from Monte Carlo $\Delta_{o,\text{MC}} = 1.2751(6)$. The lowest $\mathbb{Z}_2$-even primary has scaling dimension $2.9(1)$, in agreement with the theoretical expectation for the displacement operator $\Delta_D = 3$. Moreover, we verify that their descendants $\partial o, \partial\bar{\partial}o, \partial^2 o$ and $\partial D$ with $l_z = 1, 2, 0$ and 1 scale to integer spacing with $\Delta_o$ and $\Delta_D$ in the thermodynamic limit.

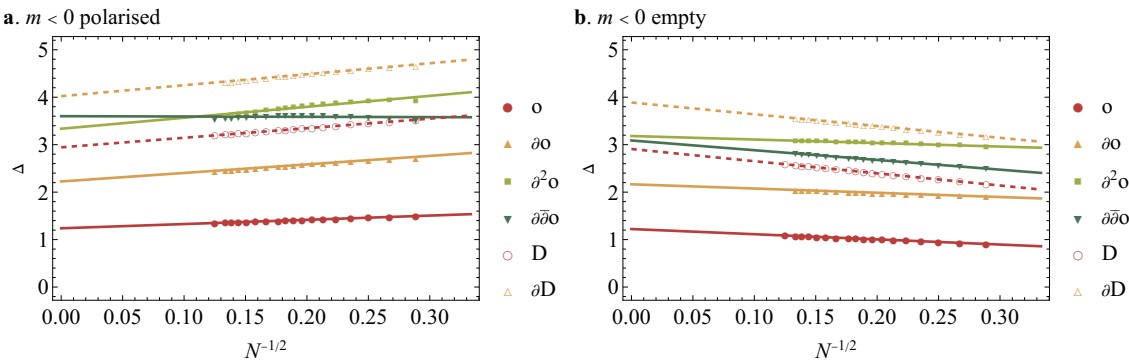

Figure 3: The scaling dimensions of various operators for system size $12 \leq N \leq 64$ in the orbital space boundary condition that (a) the $m < 0$ orbitals are half-filled polarised towards $+x$ direction and (b) the $m < 0$ orbitals are set to be empty. The two realisations agree in the thermodynamic limit.

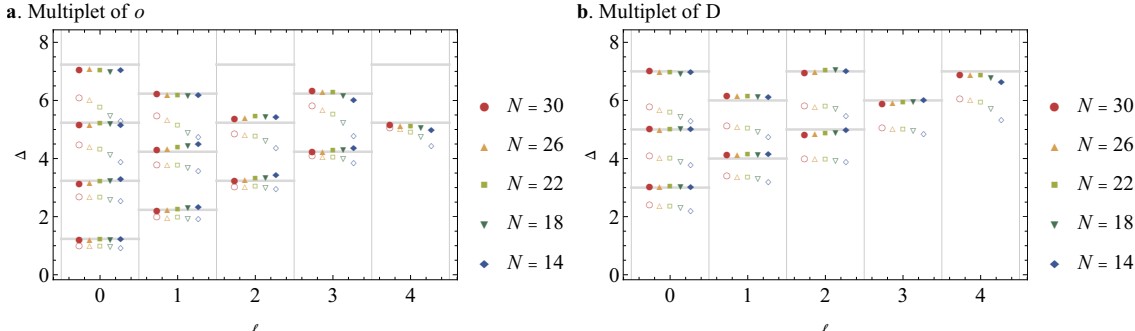

Figure 4: The scaling dimensions of the conformal multiplets of (a) $o$ and (b) D for system sizes $14 \leq N \leq 30$ in the orbital space boundary condition that the $m < 0$ orbitals are empty. The solid symbols represent the result after taking into account the correction by the irrelevant boundary operator D, and the empty symbols represent the raw data.

In addition, we can identify more descendants of $o$ and D in the energy spectrum. In Figure 4

(and also Table 3 in Appendix C.2), we identified all their descendant operators up to $\Delta = 7$ and $l_z = 4$. The raw data obtained directly from Eq. (10) (empty symbols in Figure 4) suffer from strong finite-size corrections. The finite-size corrections can be systematically removed by taking into account irrelevant operators in conformal perturbation theory [49] (*cf.* Appendix B for details). After this procedure (solid symbols in Fig. 4), we find that all their scaling dimensions are in agreement with the integer spacing up to a maximal discrepancy of 2% at the size $N = 30$.

### 3.2 Spectrum of normal surface CFT

We then turn to the spectrum of normal surface CFT. Lattice simulation [26] and conformal bootstrap [24] have reported the lowest primary operator being the displacement operator at $\Delta_D = 3$.

On fuzzy sphere, we realise the normal surface CFT by applying the boundary condition that $m < 0$ orbitals are half-filled and polarised towards $z$-direction. The finite size scaling of the scaling dimensions of the lowest operators up to $N^{-3/2}$ order are plotted in Figure 5a and listed in Table 4 in the Appendix C.2. We estimate that the lowest primary has scaling dimension 2.91(25), in agreement with the theoretical expectation for the displacement operator $\Delta_D = 3$. Its descendants $\partial D, \partial \bar{\partial} D$ and $\partial^2 D$ scale to integer spacing in the thermodynamic limit. We also identified all the descendant operators of D up to $\Delta = 7$ and $l_z = 4$ in Figure 5b. After removing the contributions from the leading irrelevant surface operator D from conformal perturbation theory, we find that all their scaling dimensions are in agreement with the integer spacing up to a maximal discrepancy of 1% at the size $N = 30$.

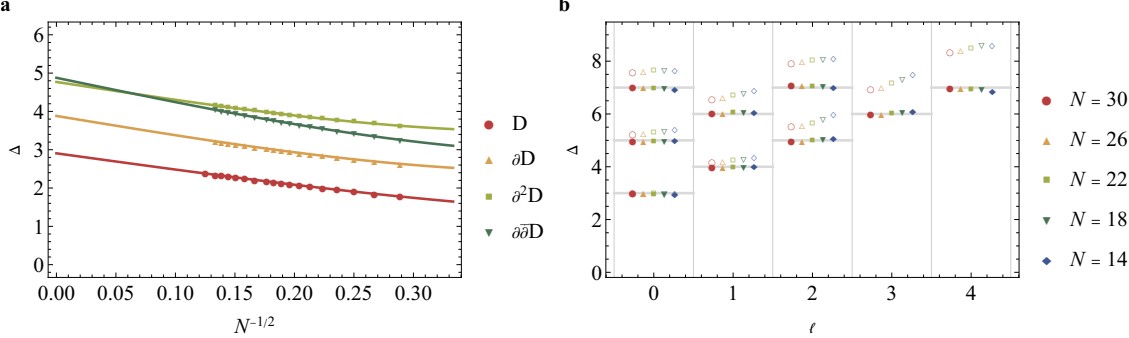

Figure 5: (a) The scaling dimensions of various operators calibrated by the bulk stress tensor $T^{\mu\nu}$ for system size $12 \le N \le 64$ in normal surface CFT realised by the orbital space boundary condition that the $m < 0$ orbitals are half-filled polarised towards $+z$ direction. (b) The scaling dimensions of the conformal multiplets of D in the same setup for $N \le 30$. The solid symbols represent the result after taking into account the correction by the irrelevant boundary operator D, and the empty symbols represent the raw data.

# 4  Orbital space boundary scheme: Correlation functions

We now turn to another important piece of data of the surface CFTs, the OPE coefficients which determine correlation functions. The simplest correlators in surface CFTs are the bulk one-point function and bulk-to-surface two-point function

$$G_\phi(x) = \langle \phi(x) \rangle = \frac{a_\phi}{|2x_\perp|^{\Delta_\phi}}$$

$$G_{\phi\hat\phi}(x) = \langle \phi(x)\hat\phi(0) \rangle = \frac{b_{\phi\hat\phi}}{|2x_\perp|^{\Delta_\phi - \Delta_{\hat\phi}}|x|^{2\Delta_{\hat\phi}}}, \tag{13}$$

where $\phi$ is a bulk operator and $\hat\phi$ is a surface operator. The $a_\phi$ and $b_{\phi\hat\phi}$ are universal OPE coefficients from bulk to surface identity or a nontrivial surface operator. Making use of the state-operator correspondence and the Weyl transformation, the two-point function can be written as an inner product on the sphere

$$G_{\phi\hat\phi}(x) = R^{\Delta_\phi}\langle \hat{0}|\phi(\Omega)|\hat\phi\rangle = b_{\phi\hat\phi}/2^{\Delta_\phi + \Delta_{\hat\phi}}(\cos\theta)^{\Delta_\phi - \Delta_{\hat\phi}} \tag{14}$$

where $R = |x|$, $\Omega$ denotes the direction of $x$ and $|\hat\phi\rangle$ is the state corresponding to surface operator $\hat\phi$.

On the fuzzy sphere, the simplest local bulk observable is the density operator

$$n^i(\Omega) = \mathbf{\Psi}^\dagger(\Omega)\sigma^i\mathbf{\Psi}(\Omega) = \sum_{mm'} Y_{sm}^{(s)}(\Omega)\bar{Y}_{sm'}^{(s)}(\Omega)\mathbf{c}_m^\dagger\sigma^i\mathbf{c}_{m'} \tag{15}$$

where $i = x, z$. From the CFT perspective, the density operators are the superpositions of scaling operators with corresponding quantum numbers. In the leading order, they can be used as UV realisations of CFT operators $\sigma$ and $\epsilon$.

$$n^x = \lambda_0 + \lambda_\epsilon R^{-\Delta_\epsilon}\epsilon + \lambda_{\partial_\tau\epsilon}R^{-\Delta_\epsilon - 1}\partial_\tau\epsilon + \lambda_{T_{\tau\tau}}R^{-3}T_{\tau\tau} + \dots \qquad \epsilon_{\rm FS} = \frac{n^x - \lambda_0}{\lambda_\epsilon R^{-\Delta_\epsilon}}$$

$$n^z = \lambda_\sigma R^{-\Delta_\sigma}\sigma + \lambda_{\partial_\tau\sigma}R^{-\Delta_\sigma - 1}\partial_\tau\epsilon + \lambda_{\partial^2\sigma}R^{-\Delta_\sigma - 2}\partial^2\sigma + \dots \qquad \sigma_{\rm FS} = \frac{n^z}{\lambda_\sigma R^{-\Delta_\sigma}} \tag{16}$$

where UV-dependent coefficients $\lambda_0, \lambda_\sigma, \lambda_\epsilon$ can be computed from the bulk correlation function without the boundary. The subleading terms to the density operators contribute to the finite size corrections

$$\epsilon_{\rm FS} = \epsilon + \frac{\lambda_{\partial_\tau\epsilon}}{\lambda_\epsilon}R^{-1}\partial_\tau\epsilon + \frac{\lambda_{T_{\tau\tau}}}{\lambda_\epsilon}R^{-3+\Delta_\epsilon}T_{\tau\tau} + \dots$$

$$\sigma_{\rm FS} = \sigma + \frac{\lambda_{\partial_\tau\sigma}}{\lambda_\sigma}R^{-1}\partial_\tau\sigma + \frac{\lambda_{\partial^2\sigma}}{\lambda_\sigma}R^{-2}\partial^2\sigma + \dots \tag{17}$$

where $R = N^{1/2}$ in the fuzzy sphere. Another source of finite size effect is the correction to the states $|\hat\phi\rangle$ by irrelevant surface operator D. Its contribution should scale with $R^{2-\Delta_D} = N^{-1/2}$.

## 4.1  Correlation functions of ordinary surface CFT

In the ordinary surface CFT, we calculate the one-point function $G_\epsilon$ and the two-point functions $G_{\epsilon D}$ and $G_{\sigma o}$ (Figure 6). All the correlators approach the CFT predictions with increasing system

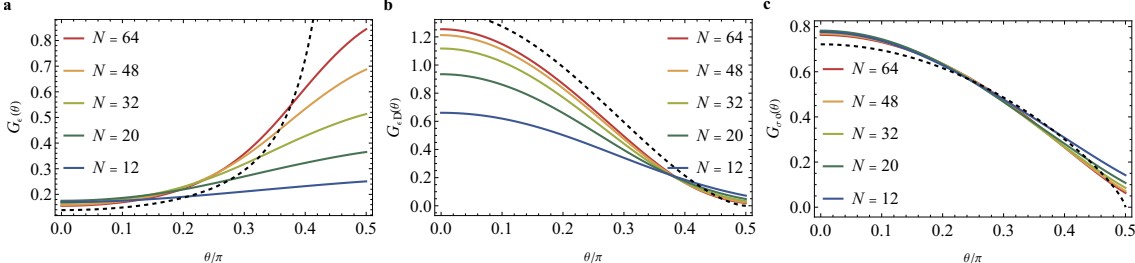

Figure 6: The one-point function (a) $G_\epsilon$ and the two-point functions (b) $G_{\epsilon D}$, (c) $G_{\sigma o}$ in the ordinary surface CFT as a function of angle $\theta$ at finite size calculated in the orbital space boundary condition that the $m < 0$ orbitals are half-filled and polarised in $x$-direction, compared with the prediction by conformal symmetry (black dashed line).

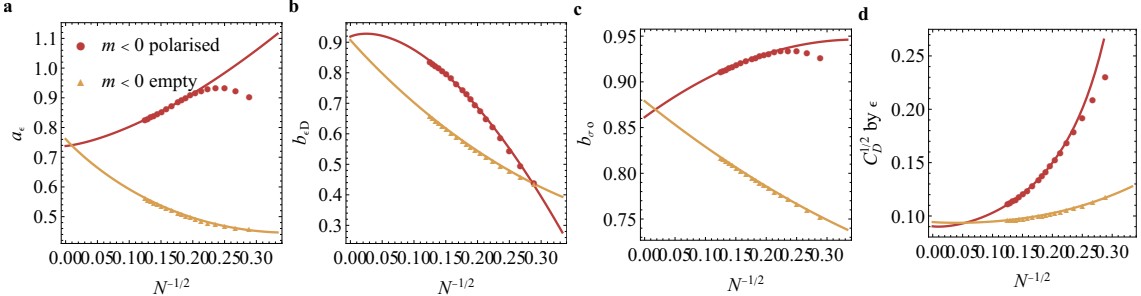

Figure 7: The finite-size extrapolation of the one-point and two-point OPE coefficients (a) $a_\epsilon$, (b) $b_{\epsilon D}$, (c) $b_{\sigma o}$ and the (d) Zamolodchikov norm $C_D^{1/2}$ calculated from $\epsilon$ in the ordinary surface CFT both in the orbital space boundary condition that the $m < 0$ orbitals are half-filled and polarised in $x$-direction and that they are empty.

size. In particular, the correlation function at $\theta = \pi/2$ increases with the system size, corresponding to the UV divergence of the one-point correlation function at $x_\perp = 0$. We further analyze these correlation functions quantitatively in Appendix C.3.

The OPE coefficients $a_\epsilon$, $b_{\epsilon D}$ and $b_{\sigma o}$ can also be extracted from the values of the correlation functions at the north pole $G(+\hat{z})$. We calculate the finite size values both for the boundary conditions that the $m < 0$ orbitals are empty and for $m < 0$ orbitals are half-filled and polarised to $+x$-direction and carry out a finite size scaling according to Eq. (17) (Figure 7). The results from both schemes agree in the thermodynamic limit, and the one-point function $a_\epsilon$ agrees with the conformal bootstrap result.

Another interesting quantity that we can extract is the Zamolodchikov norm $C_D$, which is defined by the two-point function of the displacement operator[5] [4]

$$\langle \tilde{D}(x)\tilde{D}(0) \rangle = C_D/x^6. \tag{18}$$

The displacement operator is the perpendicular component of the divergence of bulk stress tensor on the boundary, so the value of $C_D$ measures the breaking of translational symmetry. For any bulk operator[6], the 1-pt function $\langle \phi(x) \rangle = a_\phi/|2x_\perp|^\Delta$ and the 2-pt function

---

[5]Here we use $\tilde{D}$ to denote the displacement operator in the normalisation of Ref [4], and D to denote our normalisation

[6]In this paper, we restrict ourselves to Lorentz scalars.

$\langle \phi(x)\tilde{D}(0)\rangle = b_{\phi\tilde{D}}/(|2x_\perp|^{\Delta_\phi-3}|x|^6)$ are related by a Ward identity $\Delta a_\phi/b_{\phi\tilde{D}} = 4\pi$ [4]. In this paper, we adopt a convention such that the two-point function of the displacement operator is normalised $\langle D(x)D(0)\rangle = 1/x^6$. In this convention, the Zamolodchikov norm $C_D$ can be determined by the Ward identity

$$C_D^{1/2} = \frac{1}{4\pi}\frac{\Delta_\phi a_\phi}{b_{\phi D}} \tag{19}$$

where $\phi$ is an arbitrary bulk operator with non-vanishing $a_\phi$. We extract this quantity by choosing the bulk operator $\epsilon$.

Our results are listed in the upper panel of Table 2. We note that the OPE coefficient $b_{\epsilon D}$ is not currently available by conformal bootstrap and is reported for the first time.

Table 2: The one-point and two-point OPE coefficient and the Zamolodchikov norm $C_D$ calculated in the fuzzy sphere in the orbital space boundary scheme compared with the bootstrap result. For the ordinary surface CFT, the two lines are calculated in the boundary condition where the $m < 0$ orbitals are empty and polarised to $x$-direction respectively. The 2-pt OPE $b_{\epsilon D}$ is not available by conformal bootstrap and are reported for the first time in our paper. The error bars are estimated from finite-size extrapolation.

| Correlator | | Previous | Fuzzy sphere |
|---|---|---|---|
| Ordinary | $a_\epsilon$ | 0.750(3) [24] | $0.74(4)\ \ +0.17N^{-1/2}+1.84N^{-0.79}$ |
| | | | $0.76(10)-2.41N^{-1/2}+2.80N^{-0.79}$ |
| | $b_{\epsilon D}$ | N/A | $0.92(4)\ \ +1.02N^{-1/2}-5.62N^{-0.79}$ |
| | | | $0.91(12)-2.39N^{-1/2}+1.18N^{-0.79}$ |
| | $b_{\sigma o}$ | 0.755(13) | $0.86(2)\ \ +0.49N^{-1/2}-0.69N^{-1}$ |
| | | | $0.88(3)\ \ -0.54N^{-1/2}+0.33N^{-1}$ |
| | $C_D$ by $\epsilon$ | N/A | 0.0082(19) |
| | | | 0.0089(2) |
| Normal | $a_\epsilon$ | 6.607(7) [24] | $6.4(9)\ \ \ \ -19.5N^{-1/2}+19.9N^{-0.79}$ |
| | $a_\sigma$ | 2.599(1) [24], 2.60(5) [18] | $2.59(16)-2.81N^{-1/2}+1.76N^{-1}$ |
| | $b_{\epsilon D}$ | 1.742(6) [24] | $1.74(22)-4.86N^{-1/2}+4.66N^{-0.79}$ |
| | $b_{\sigma D}$ | 0.25064(6) [24], 0.244(8) [18] | $0.254(17)-0.30N^{-1/2}+0.20N^{-1}$ |
| | $C_D$ by $\epsilon$ | 0.182(1) [24] | 0.174(3) |
| | $C_D$ by $\sigma$ | 0.182(1) [24], 0.193(5) [18] | 0.177(2) |

## 4.2 Correlation functions of normal surface CFT

In the normal surface CFT, we calculate the one-point function $G_\epsilon, G_\sigma$ and the two-point functions $G_{\epsilon D}, G_{\sigma D}$ (Figure 8). All the correlators approach the CFT predictions with increasing system sizes.

We extract the OPE coefficients $a_\epsilon, a_\sigma, b_{\epsilon D}$ and $b_{\sigma D}$ (Figure 9) and find that they agree with the conformal bootstrap result. Another non-trivial cross-check is the Ward identity that the Zamolodchikov norm $C_D$ calculated from different bulk operators $\phi$ should all be the same. We

extract $C_D$ values by $\sigma$ and $\epsilon$ and find that they agree well. Our results are listed in the lower panel of Table 2.

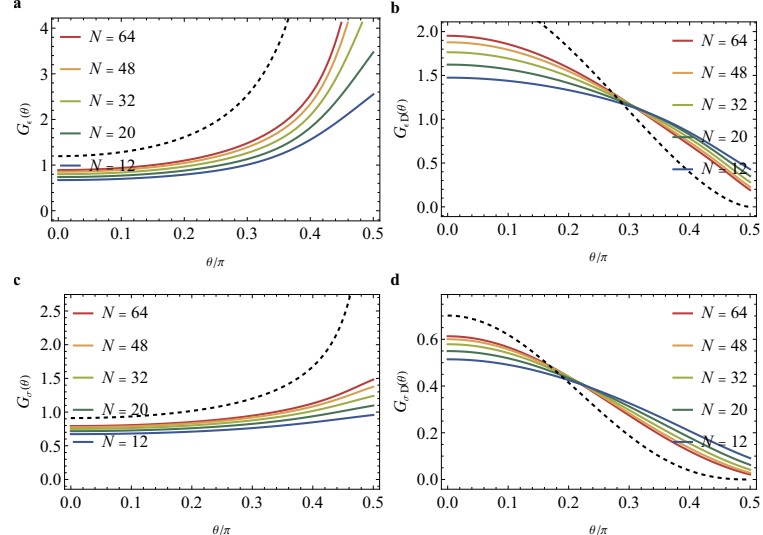

Figure 8: The one-point and two-point functions (a) $G_\epsilon$, (b) $G_{\epsilon D}$, (c) $G_\sigma$ and (d) $G_{\sigma D}$ in the normal surface CFT as a function of angle $\theta$ at finite size calculated in the orbital space boundary scheme, compared with the prediction by conformal symmetry (black dashed line).

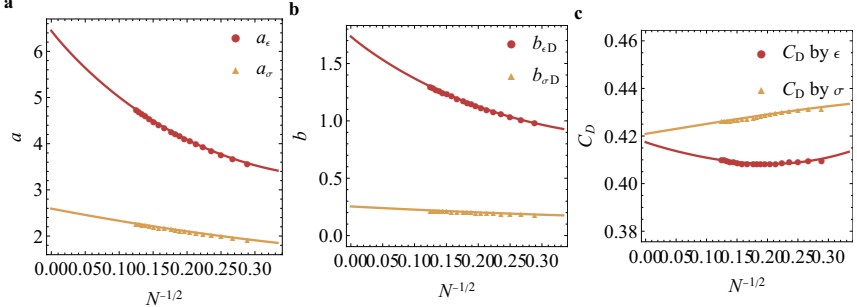

Figure 9: The finite-size extrapolation of the (a) one-point OPE coefficients $a_\epsilon$, $a_\sigma$, (b) two-point OPE coefficients $b_{\sigma D}$, $b_{\epsilon D}$, and (c) the Zamolodchikov norm $C_D^{1/2}$ calculated from $\epsilon$ and $\sigma$ in the normal surface CFT in the orbital space boundary scheme.

# 5  Real space boundary scheme

We now turn to the real space boundary scheme. We keep the boundary pinning field $h_s$ the same for different sizes during finite-size scaling. We confirm that the scaling dimensions and correlation functions do not depend on $h_s$ in the thermodynamic limit.

We first calculate the scaling dimensions of the lowest-lying operators. We find that the scaling dimensions scale to the same value as in the orbital space boundary scheme both in the ordinary

and the normal surface CFTs (Figure 10).

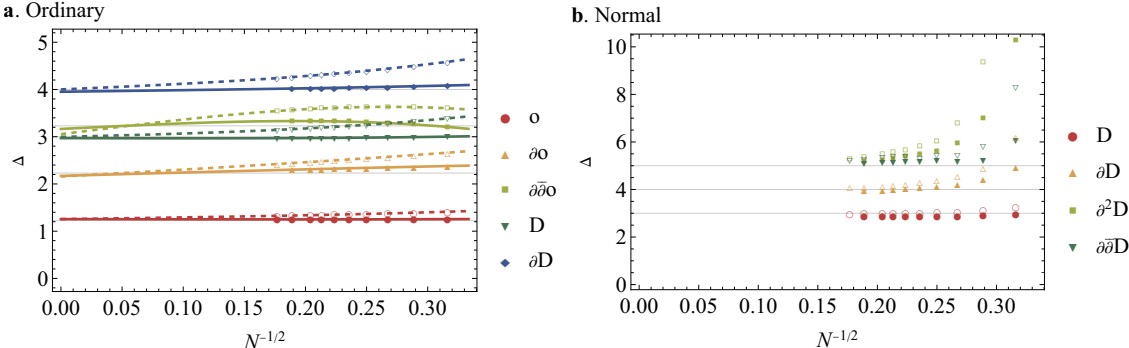

Figure 10: The scaling dimensions of various operators for system size $12 \leq N \leq 32$ for the (a) ordinary and (b) normal surface CFTs in the real space boundary scheme. In panel (a), we set $h_s = 10$ for the solid lines and symbols and $h_s = 20$ for the dashed lines and empty symbols; In panel (b), we set $h_s = 500$ for the solid symbols and $h_s = 1000$ for the empty symbols.

We have also repeated the calculation for the one-point and two-point functions both in the ordinary and the normal surface CFTs. We confirm that all these correlation functions approach the CFT predictions as the system size increases (Figure 11, and Figures 18 and 19 in Appendix C.4). In the small $\theta$ region (far away from the boundary), we find particularly good agreement.

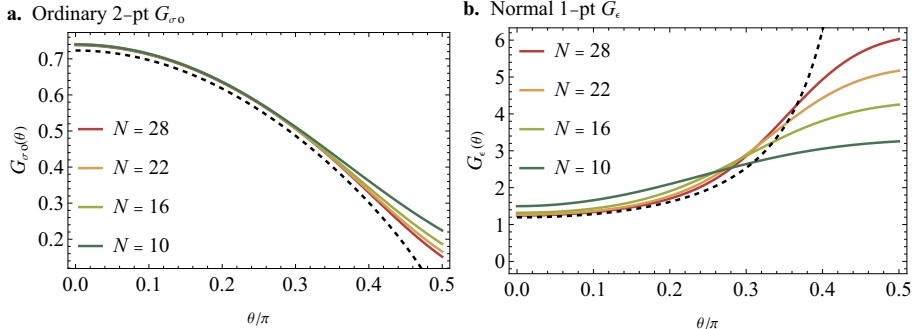

Figure 11: (a) The two-point function $G_{\sigma o}$ in the ordinary surface CFT and (b) the one-point function $G_\epsilon$ in the normal surface CFT as a function of angle $\theta$ at finite size calculated in the real space boundary scheme, compared with the prediction by conformal symmetry (black dashed line). We have set $h_s = 10$ for (a) and $h_s = 500$ for (b).

# 6  Boundary central charge and OPE from wavefunction overlap

In this section, we present an alternative approach to calculate the conformal data of surface CFT by taking the overlap of the bulk eigenstates with a polarised bulk state [35, 50–52]. We consider the overlap between a CFT state $|\phi\rangle$ and a polarised state, e.g., $|\hat{\mathbf{z}}\rangle = \prod_m c^{\dagger}_{m,\uparrow}|\text{vac}\rangle$ and $|\hat{\mathbf{x}}\rangle = \prod_m \frac{1}{\sqrt{2}}(c^{\dagger}_{m,\uparrow} + c^{\dagger}_{m,\downarrow})|\text{vac}\rangle$. The overlap can be represented as the path integral on the

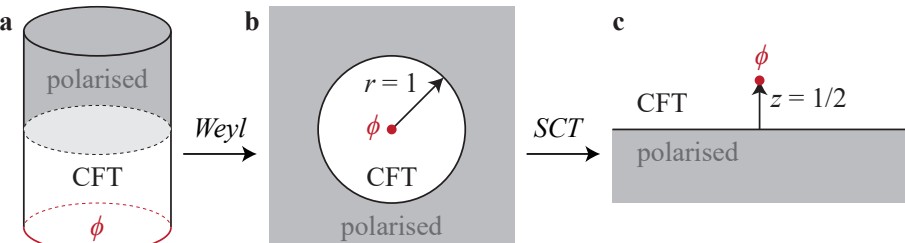

Figure 12: An illustration of transforming the overlap between CFT state $|\phi\rangle$ and polarised state. (a) The path integral configuration of the overlap between CFT state $|\phi\rangle$ and polarised state. (b) Through Weyl transformation, the overlap is equivalent to the CFT configuration within a sphere $r < 1$ on flat spacetime and an operator $\phi$ inserted at the centre, and the boundary condition is determined by the polarised state. (c) Through special conformal transformation, the overlap is equivalent to the CFT configuration on half infinite spacetime $z > 0$ and an operator inserted at $z = 1/2$.

cylinder $S^2 \times \mathbb{R}_+$ (Figure 12a), where $\tau = 0$ corresponds to the boundary of the CFT. When the polarised state is taken as $|\hat{\mathbf{x}}\rangle$, the path integral corresponds to an ordinary surface CFT; when the polarised state is taken as $|\hat{\mathbf{z}}\rangle$, the path integral corresponds to a normal surface CFT. Through a Weyl transformation, the interface on the cylinder is transformed into the unit sphere on the flat spacetime (Figure 12b) with the CFT living in the interior, and the state $|\phi\rangle$ corresponds to a bulk operator $\phi$ inserted at the centre of the sphere.

When $\phi$ is taken as the identity operator, the scaling behaviour of $\langle \text{pol}|\mathbb{I}\rangle$ is related to the boundary central charge $c_{\text{bd}}$. The boundary central charge is defined through the response of surface CFTs to gravity [13, 14, 53, 54].

$$T_\mu{}^\mu = \frac{\delta(x_\perp)}{4\pi}(c_{\text{bd}}\hat{R} + b \operatorname{tr}\hat{K}^2) \tag{20}$$

where $\hat{R}$ is the Ricci scalar, and $\hat{K}$ is the traceless part of the extrinsic curvature associated with the boundary. The value of universal property $b$ is related to the Zamolodchikov norm $b = \frac{\pi^2}{8}C_{\text{D}}$ [54], whereas the non-perturbative calculation of the boundary central charge $c_{\text{bd}}$ is previously unknown.

We show that $c_{\text{bd}}$ can be extracted from the scaling behaviour of $\langle \text{pol}|\mathbb{I}\rangle$. The overlap $\langle \text{pol}|\mathbb{I}\rangle$ is proportional to the partition function of surface CFT on 2-sphere, where $\hat{R} = 2/r^2$ and $K_{ab} = 0$. Its scaling behaviour is given by [53, 55]

$$\begin{aligned}
\log Z &= \frac{\log r\mu}{24\pi} \int \mathrm{d}^2\sigma \sqrt{\hat{g}}(c_{\text{bd}}\hat{R} + b\hat{K}_{ab}\hat{K}^{ab}) + (\text{divergent}) \\
&= \frac{c_{\text{bd}}}{3}\log(r\mu) + (\text{divergent})
\end{aligned} \tag{21}$$

where the divergent term is proportional to $r^2$. We find that $c_{\text{bd}}^{(\text{O})} = -0.0159(5)$ for ordinary surface CFT and $c_{\text{bd}}^{(\text{N})} = -1.44(6)$ for normal surface CFT. This is the first time the boundary central charges are computed in a non-perturbative way.

On the other hand, the ratios of the overlap on the cylinder between the excited and ground states correspond to the one-point correlation function of a bulk operator

$$\frac{\langle \text{pol}|\phi\rangle}{\langle \text{pol}|\mathbb{I}\rangle} = \langle\phi(0)\rangle_{\text{flat, interface at } r=1} \tag{22}$$

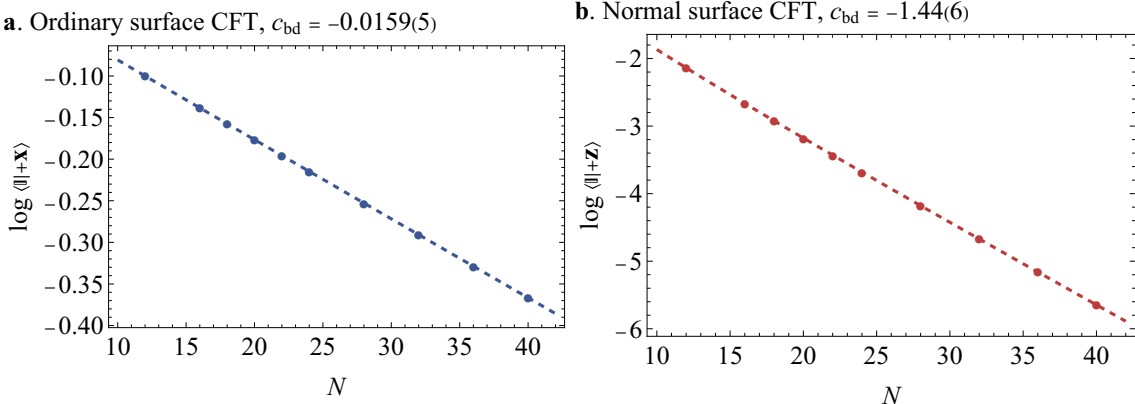

Figure 13: The finite size scaling of the overlaps of the bulk Ising ground state and the (a) $x$ polarised state, (b) $z$ polarised state that gives the boundary central charge of (a) ordinary and (b) normal surface CFT. The fitting ansatz is taken as $\log\langle \mathbb{I}|\text{pol}\rangle = C_1 N + C_0 + C_{-1}/N + (c_{\text{bd}}/6)\log N$. The error bar is estimated from finite-size extrapolation, where the finite-size $c_{\text{bd}}$ is estimated by taking the derivative $c_{\text{bd}}(N) = -(N^2/6)(\partial^2/\partial N^2)\log\langle \mathbb{I}|\text{pol}\rangle(N)$

where the denominator is taken as the overlap between the polarised state and the CFT ground state to ensure proper normalisation. Further using a special conformal transformation (SCT)

$$\mathbf{x} \mapsto \frac{\mathbf{x} - \mathbf{b}x^2}{1 - 2\mathbf{x}\cdot\mathbf{b} + b^2 x^2} \text{ with } \mathbf{b} = \hat{\mathbf{z}} \tag{23}$$

the unit sphere is transformed into the plane $z = -1/2$, and then after a translation, we recover the standard setup where the surface situates on $z = 0$ plane, and the operator is placed at $\hat{\mathbf{z}}/2$. The one-point correlator is converted into the standard one-point correlation function in the surface CFT, which exactly equals $a_\phi$,

$$\frac{\langle\text{pol}|\phi\rangle}{\langle\text{pol}|\mathbb{I}\rangle} = a_\phi. \tag{24}$$

We note that the SCT does not contribute a conformal factor for operators at the origin.

Using this method, we are able to calculate the one-point OPE coefficients $a_\epsilon$ in ordinary surface CFT and $a_\sigma, a_\epsilon$ in normal surface CFT (Figure 14). The results are consistent with previous sections where we realise the boundary in real space. It is worth noting that this method does not involve realising the CFT operators with density operators on the fuzzy sphere. Thus, the method can also be used to compute the one-point OPE coefficients of higher bulk operators such as $\sigma'$ and $\epsilon'$, and is subject to one fewer source of finite size corrections.

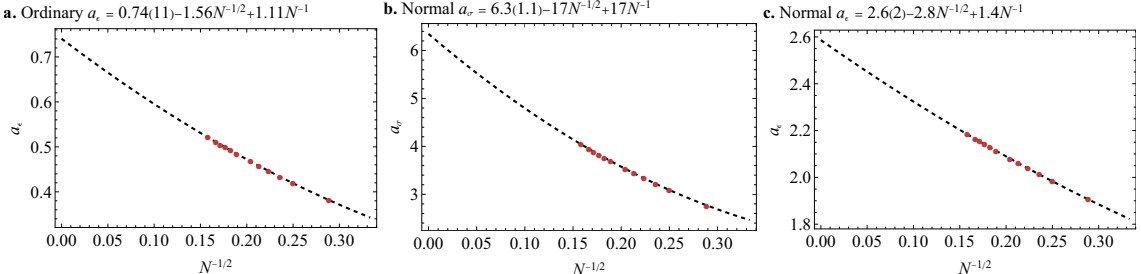

Figure 14: The finite size scaling of one-point OPE coefficients (a) $a_\epsilon$ in ordinary surface CFT and (b) $a_\sigma$, (c) $a_\epsilon$ in normal surface CFT calculated by taking the overlap between the bulk Ising CFT state and a bulk direct product state.

## 7   Discussion and summary

In this paper, we have studied surface critical behaviours in the 3d Ising CFT with the setup of fuzzy sphere. We realise the ordinary and normal surface CFTs through the real-space scheme that adds a real space pinning field to remove the degrees of freedom on the southern hemisphere, and the orbital-space scheme that pins the electrons on the $m < 0$ orbitals. We have studied the operator spectra, and identified the surface primary $o$ in the ordinary surface CFT for which $\Delta_o = 1.23(4)$ by state-operator correspondence, the displacement operator D in both surface CFTs and their descendants up to $\Delta \leq 7, l_z \leq 4$. We find their scaling dimensions agree perfectly with the prediction of conformal symmetry and also match previous results obtained by conformal bootstrap and Monte Carlo after a finite size analysis. We have also studied the one-point and two-point correlation functions. The correlation functions calculated in fuzzy sphere approach the prediction of conformal symmetry with increasing system size. We have also calculated the bulk one-point and bulk-to-surface two-point OPE coefficients. Our results match the results of conformal bootstrap and also agree with the prediction of Ward identity. Some of the 2-pt functions such as $b_{\sigma o}$ and $b_{\epsilon D}$ are reported for the first time. We have further presented an alternative approach to calculate the conformal data of surface CFT by measuring the overlap between a bulk CFT state and a polarised state. In particular, we report for the first time the non-perturbative result of the boundary central charges $c_{\text{bd}}^{(\text{O})} = -0.0159(5)$ and $c_{\text{bd}}^{(\text{N})} = -1.44(6)$.

In the orbital space boundary scheme, removing the pinned orbitals can be realised by adding a magnetic field on the remaining orbitals that is proportional to the polarisation $n_{m<}^\alpha = \langle \mathbf{c}_{m<}^\dagger \sigma^\alpha \mathbf{c}_{m<} \rangle$ of the removed orbital $m^<$, where $\alpha = 0, x, y, z$. The polarisation vector $n_{m<}^\alpha$ serves as tuning parameters in the surface CFT. In the paper, we have shown that both $n_{m<}^x = 0$ and $n_{m<}^x = 1$ correspond to the ordinary surface CFT. The choice $n_{m<}^x = -1$ leads to a spontaneous symmetry breaking on the surface and is likely to correspond to the extraordinary surface CFT. By fine-tuning $-1 < n_{m<}^x < 0$, one would find a surface critical point that is likely to correspond to the special surface CFT; by fine-tuning $0 < n_{m<}^x < 1$, one is likely to find a parameter where the leading irrelevant surface perturbation D of the ordinary surface CFT is tuned away, which would reduce finite-size corrections to the spectrum and correlators as well as higher surface primaries and their multiplets. Starting from the extraordinary surface CFT, one can reach the surface spontaneous

symmetry-breaking phase transition by increasing the bulk transverse field. One may further study the RG flow between the 3d surface criticality and the pure 2d surface transition.

The fuzzy sphere can be used to study various other conformal defects and other conformal field theories. For example, by adding a real space magnetic field at the equator, one can realise a plane defect in the spacetime. One may alternatively set the orbital space boundary cut at $m_c(\theta) = (s+1)(1 + \cos\theta/2)/2$ instead of $m_c = 0$ and realise a cone defect. Moreover, one can consider the coexistence of multiple kinds of defects, *e.g.*, the setup where a half-infinite line defect ends at a surface, or where the line defect lives on a surface. The fusion of different defects can also be studied on the fuzzy sphere, see Ref. [56] for a study of the fusion of line defects. One may generalise it to higher dimensional defects such as boundaries or interfaces. Finally, starting from other bulk CFTs on fuzzy sphere such as O($N$) Wilson-Fisher (see Ref. [57] for a recent theoretical exploration) or SO(5) DQCP, one can also study a richer set of surface critical behaviours which keep different subgroups of the bulk global symmetries.

## Acknowledgments

We would like to thank Yin-Chen He, Zohar Komargodski, Max Metlitski, Francesco Parisen Toldin, Marco Meineri, Mykola Dedushenko, Ryan Lanzetta, Wei Zhu for illuminating discussions. Part of the numerical calculations are done using the packages FuzzifiED and ITensor. Z.Z. acknowledges support from the Natural Sciences and Engineering Research Council of Canada (NSERC) through Discovery Grants. Research at Perimeter Institute is supported in part by the Government of Canada through the Department of Innovation, Science and Industry Canada and by the Province of Ontario through the Ministry of Colleges and Universities.

*Note: While preparing this manuscript, we became aware of a similar work [58] by Mykola Dedushenko studying the Ising surface CFTs by fuzzy sphere.*

## A  Effective Hamiltonian after pinning orbitals

We divide all the orbitals $m = -s, \ldots, s$ into two parts, namely $m^>$ to be kept and $m^<$ to be removed. We want to rewrite the Hamiltonian

$$H = \sum_{m_1 m_2 m_3 m_4} V_{m_1 m_2 m_3 m_4} c^\dagger_{m_1 \downarrow} c^\dagger_{m_2 \uparrow} c_{m_3 \uparrow} c_{m_4 \downarrow} - h \sum_m \mathbf{c}^\dagger_m \sigma^x \mathbf{c}_m \tag{25}$$

into $H^>$ that contains only $c^{(\dagger)}_{m^>}$. Due to the conservation of particle numbers at a removed orbital, any $c_{m^<}$ must be paired with a $c^\dagger_{m^<}$ with the same $m^<$ in a four-fermion term. Hence, any four-fermion term must contain 0, 2 or 4 removed orbital, and the four-fermion terms with all the orbitals removed become a constant number that can be dropped.

$$H^> = \sum_{m_1^> m_2^> m_3^> m_4^>} V_{m_1^> m_2^> m_3^> m_4^>} c^\dagger_{m_1^> \downarrow} c^\dagger_{m_2^> \uparrow} c_{m_3^> \uparrow} c_{m_4^> \downarrow} - h \sum_{m^>} \mathbf{c}^\dagger_{m^>} \sigma^x \mathbf{c}_{m^>}$$

$$+ \sum_{m^> m^<} V_{m^> m^< m^< m^>} (c^\dagger_{m^> \downarrow} c^\dagger_{m^< \uparrow} c_{m^< \uparrow} c_{m^> \downarrow} + c^\dagger_{m^< \downarrow} c^\dagger_{m^> \uparrow} c_{m^> \uparrow} c_{m^< \downarrow}$$

$$- c^\dagger_{m^> \downarrow} c^\dagger_{m^< \uparrow} c_{m^> \uparrow} c_{m^< \downarrow} - c^\dagger_{m^< \downarrow} c^\dagger_{m^> \uparrow} c_{m^< \uparrow} c_{m^> \downarrow})$$

$$= \sum_{m_1^> m_2^> m_3^> m_4^>} V_{m_1^> m_2^> m_3^> m_4^>} c^\dagger_{m_1^> \downarrow} c^\dagger_{m_2^> \uparrow} c_{m_3^> \uparrow} c_{m_4^> \downarrow} - h \sum_{m^>} \mathbf{c}^\dagger_{m^>} \sigma^x \mathbf{c}_{m^>}$$

$$+ \sum_{m^> m^<} V_{m^> m^< m^< m^>} (c^\dagger_{m^> \downarrow} c_{m^> \downarrow} \langle c^\dagger_{m^< \uparrow} c_{m^< \uparrow} \rangle + c^\dagger_{m^> \uparrow} c_{m^> \uparrow} \langle c^\dagger_{m^< \downarrow} c_{m^< \downarrow} \rangle$$

$$+ c^\dagger_{m^> \downarrow} c_{m^> \uparrow} \langle c^\dagger_{m^< \uparrow} c_{m^< \downarrow} \rangle + c^\dagger_{m^> \uparrow} c_{m^> \downarrow} \langle c^\dagger_{m^< \downarrow} c_{m^< \uparrow} \rangle)$$

$$= \sum_{m_1^> m_2^> m_3^> m_4^>} V_{m_1^> m_2^> m_3^> m_4^>} c^\dagger_{m_1^> \downarrow} c^\dagger_{m_2^> \uparrow} c_{m_3^> \uparrow} c_{m_4^> \downarrow} - h \sum_{m^>} \mathbf{c}^\dagger_{m^>} \sigma^x \mathbf{c}_{m^>}$$

$$+ \sum_{m^> m^<} V_{m^> m^< m^< m^>} (c^\dagger_{m^> \downarrow} c_{m^> \downarrow} \frac{n^0_{m^<} + n^z_{m^<}}{2} + c^\dagger_{m^> \uparrow} c_{m^> \uparrow} \frac{n^0_{m^<} - n^z_{m^<}}{2}$$

$$+ c^\dagger_{m^> \downarrow} c_{m^> \uparrow} \frac{n^x_{m^<} + i n^y_{m^<}}{2} + c^\dagger_{m^> \uparrow} c_{m^> \downarrow} \frac{n^x_{m^<} - i n^y_{m^<}}{2})$$

$$= \sum_{m_1^> m_2^> m_3^> m_4^>} V_{m_1^> m_2^> m_3^> m_4^>} - h \sum_{m^>} \mathbf{c}^\dagger_{m^>} \sigma^x \mathbf{c}_{m^>} + \sum_{m^>} \sum_{\alpha=0,x,y,z} h^\alpha_{m^>} \mathbf{c}^\dagger_{m^>} \sigma^\alpha \mathbf{c}_{m^>} \tag{26}$$

where

$$h^\alpha_{m^>} = \frac{1}{2} \sum_{m^<} V_{m^> m^< m^< m^>} n^\alpha_{m^<} \quad \text{and} \quad n^\alpha_{m^<} = \langle \mathbf{c}^\dagger_{m^<} \sigma^\alpha \mathbf{c}_{m^<} \rangle \tag{27}$$

In other words, when the orbitals are removed, they act as a polarising term to the kept orbitals. When the removed orbitals are set to be 'empty', we need to set $n^0 = 1$ to keep the particle-hole symmetry, and set the rest $n^{x,y,z} = 0$; when they are polarised along $z$-direction, $n^{0,z} = 1$ and the rest $n^{x,y} = 0$; when they are polarised along $x$-direction, $n^{0,x} = 1$ and the rest $n^{y,z} = 0$.

# B   Conformal perturbation

We consider the correction to the state $\phi$ with conformal dimensions $h, \bar{h}$ and its conformal family $\partial^m \bar{\partial}^{\bar{m}} \phi$ from the insertion of the operator $O$ with conformal dimensions $h_0, \bar{h}_0$.

$$H = H_{\text{CFT}} + \lambda_O \int_0^{2\pi} \frac{d\phi}{2\pi} O(\phi) \tag{28}$$

The state is formed by the operator insertion

$$|\partial^m \bar{\partial}^{\bar{m}} \phi\rangle = C_{\phi; m\bar{m}} \partial^m \bar{\partial}^{\bar{m}} \phi |0\rangle \tag{29}$$

The constant is determined by the normalisation

$$\langle \partial^m \bar{\partial}^{\bar{m}} \phi | \partial^m \bar{\partial}^{\bar{m}} \phi \rangle = |C_{\phi; m\bar{m}}|^2 \left\langle (\bar{\partial}^m \partial^{\bar{m}} \phi(0))^\dagger (\partial^m \bar{\partial}^{\bar{m}} \phi)(0) \right\rangle = 1 \tag{30}$$

The correction to the energy is

$$\delta E(\partial^m \bar{\partial}^{\bar{m}} \phi) = \lambda_O \int_0^{2\pi} \frac{d\phi}{2\pi} \langle \partial^m \bar{\partial}^{\bar{m}} \phi | O(\phi) | \partial^m \bar{\partial}^{\bar{m}} \phi \rangle$$

$$= \lambda_O \int_{|z_1|=1} \frac{dz_1 \, d\bar{z}_1}{2\pi} \frac{\left\langle (\bar{\partial}^m \partial^{\bar{m}} \phi(0))^\dagger (\partial^m \bar{\partial}^{\bar{m}} \phi)(0) O(z_1, \bar{z}_1) \right\rangle}{\left\langle (\bar{\partial}^m \partial^{\bar{m}} \phi(0))^\dagger (\partial^m \bar{\partial}^{\bar{m}} \phi)(0) \right\rangle} \tag{31}$$

We thus need to evaluate the 2-pt and 3-pt function

$$\left\langle (\bar{\partial}^m \partial^{\bar{m}} \phi(0))^\dagger (\partial^m \bar{\partial}^{\bar{m}} \phi)(0) \right\rangle \text{ and } \left\langle (\bar{\partial}^m \partial^{\bar{m}} \phi(0))^\dagger (\partial^m \bar{\partial}^{\bar{m}} \phi)(0) O(z_1, \bar{z}_1) \right\rangle \tag{32}$$

By definition

$$\begin{aligned}
(\bar{\partial}_w^m \partial_w^{\bar{m}} \phi(w, \bar{w}))^\dagger &= \bar{\partial}_w^m \partial_w^{\bar{m}} \left( \bar{w}^{-2h} w^{-2\bar{h}} \phi(\tfrac{1}{\bar{w}}, \tfrac{1}{w}) \right) \\
&= (-1)^{m+\bar{m}} (z^2 \partial)^m (\bar{z}^2 \bar{\partial})^{\bar{m}} \left( z^{2h} \bar{z}^{2\bar{h}} \phi(z, \bar{z}) \right)
\end{aligned} \tag{33}$$

where $w = 1/\bar{z}$. Hence (The limit $z \to \infty$, $z_0 \to 0$ and $w = 1/\bar{z} \to 0$ is assumed)

$$\begin{aligned}
\left\langle (\bar{\partial}^m \partial^{\bar{m}} \phi(0))^\dagger (\partial^m \bar{\partial}^{\bar{m}} \phi)(0) \right\rangle &= (-1)^{m+\bar{m}} (z^2 \partial)^m (\bar{z}^2 \bar{\partial})^{\bar{m}} \left( z^{2h} \bar{z}^{2\bar{h}} \partial_0^m \bar{\partial}_0^{\bar{m}} \langle \phi(z, \bar{z}) \phi(z_0, \bar{z}_0) \rangle \right) \\
&= (z^2 \partial)^m (\bar{z}^2 \bar{\partial})^{\bar{m}} \left( z^{2h} \bar{z}^{2\bar{h}} \partial^m \bar{\partial}^{\bar{m}} (z^{-2h} \bar{z}^{-2\bar{h}}) \right) \\
&= (z^2 \partial)^m (z^{2h} \partial^m z^{-2h}) \times \text{c.c.} \\
&= (-1)^m \frac{\Gamma(m + 2h)}{\Gamma(2h)} (z^2 \partial)^m (z^{-m}) \times \text{c.c.} \\
&= \frac{\Gamma(m + 2h)}{\Gamma(2h)} \bar{\partial}_w^m \bar{w}^m \times \text{c.c.} \\
&= \frac{m! \, \Gamma(m + 2h)}{\Gamma(2h)} \frac{\bar{m}! \, \Gamma(\bar{m} + 2\bar{h})}{\Gamma(2\bar{h})}
\end{aligned} \tag{34}$$

$$\begin{aligned}
&\left\langle (\bar{\partial}^m \partial^{\bar{m}} \phi(0))^\dagger (\partial^m \bar{\partial}^{\bar{m}} \phi)(0) O(z_1, \bar{z}_1) \right\rangle \\
&= (-1)^{m+\bar{m}} (z^2 \partial)^m (\bar{z}^2 \bar{\partial})^{\bar{m}} \left( z^{2h} \bar{z}^{2\bar{h}} \partial_0^m \bar{\partial}_0^{\bar{m}} \langle \phi(z, \bar{z}) \phi(z_0, \bar{z}_0) O(z_1, \bar{z}_1) \rangle \right) \\
&= f_{\phi\phi O} (-1)^{m+\bar{m}} (z^2 \partial)^m (\bar{z}^2 \bar{\partial})^{\bar{m}} \left( z^{2h} \bar{z}^{2\bar{h}} \partial_0^m \bar{\partial}_0^{\bar{m}} \left( (z - z_0)^{h_0 - 2h} (z - z_1)^{-h_0} (z_1 - z_0)^{-h_0} \times \text{c.c.} \right) \right) \\
&= f_{\phi\phi O} (-1)^m (z^2 \partial)^m \left( z^{2h} \partial_0^m \left( (z - z_0)^{h_0 - 2h} (z - z_1)^{-h_0} (z_1 - z_0)^{-h_0} \right) \right) \times \text{c.c.} \\
&= f_{\phi\phi O} \sum_{n=0}^m \binom{m}{n} \frac{\Gamma(2h - h_0 + n)}{\Gamma(2h - h_0)} \frac{\Gamma(h_0 + m - n)}{\Gamma(h_0)} \\
&\quad\quad \times (z^2 \partial)^m \left( z^{2h} (z - z_0)^{h_0 - 2h - n} (z - z_1)^{-h_0} (z_1 - z_0)^{-h_0 - m + n} \right) \times \text{c.c.} \\
&= f_{\phi\phi O} \sum_{n=0}^m \binom{m}{n} \frac{\Gamma(2h - h_0 + n)}{\Gamma(2h - h_0)} \frac{\Gamma(h_0 + m - n)}{\Gamma(h_0)} (z^2 \partial)^m \left( z^{h_0 - n} (z - z_1)^{-h_0} \right) z_1^{-h_0 - m + n} \times \text{c.c.} \\
&= f_{\phi\phi O} \sum_{n=0}^m \binom{m}{n} \frac{\Gamma(2h - h_0 + n)}{\Gamma(2h - h_0)} \frac{\Gamma(h_0 + m - n)}{\Gamma(h_0)} \bar{\partial}_w^m \left( \left( \frac{1}{\bar{w}} \right)^{h_0 - n} \left( \frac{1}{\bar{w}} - z_1 \right)^{-h_0} \right) z_1^{-h_0 - m + n} \times \text{c.c.} \\
&= f_{\phi\phi O} \sum_{n=0}^m \binom{m}{n} \frac{\Gamma(2h - h_0 + n)}{\Gamma(2h - h_0)} \frac{\Gamma(h_0 + m - n)}{\Gamma(h_0)} \bar{\partial}_w^m \left( \bar{w}^n (1 - \bar{w} z_1)^{-h_0} \right) z_1^{-h_0 - m + n} \times \text{c.c.} \\
&= f_{\phi\phi O} \sum_{n=0}^m \binom{m}{n} \frac{\Gamma(2h - h_0 + n)}{\Gamma(2h - h_0)} \frac{\Gamma(h_0 + m - n)}{\Gamma(h_0)} \\
&\quad\quad \times \binom{m}{n} (\bar{\partial}_w^n \bar{w}^n)(\bar{\partial}_w^{m-n} (1 - \bar{w} z_1)^{-h_0}) z_1^{-h_0 - m + n} \times \text{c.c.} \\
&= f_{\phi\phi O} \sum_{n=0}^m \binom{m}{n}^2 \frac{\Gamma(2h - h_0 + n)}{\Gamma(2h - h_0)} \frac{\Gamma(h_0 + m - n)}{\Gamma(h_0)}
\end{aligned}$$

$$\times n! \frac{\Gamma(h_0 + m - n)}{\Gamma(h_0)} z_1^{m-n} (1 - \bar{w} z_1)^{-h_0 - m + n} z_1^{-h_0 - m + n} \times \text{c.c.}$$

$$= f_{\phi\phi O} \sum_{n=0}^{m} \binom{m}{n}^2 n! \frac{\Gamma(2h - h_0 + n)}{\Gamma(2h - h_0)} \frac{\Gamma(h_0 + m - n)^2}{\Gamma(h_0)^2} z_1^{-h_0} \times \text{c.c.}$$

$$= f_{\phi\phi O} \frac{\Gamma(h_0 + m)^2}{\Gamma(h_0)^2} {}_3F_2(2h - h_0, -m, -m; -h_0 - m + 1, -h_0 - m + 1; 1) \times \text{c.c.} \tag{35}$$

Hence,

$$\delta E(\partial^m \bar{\partial}^{\bar{m}} \phi; \lambda_{\phi;O}, \Delta_\phi, \Delta_O)$$
$$= \lambda_{\phi;O} \frac{\Gamma(h_0 + m)^2 \Gamma(2h)}{m! \Gamma(h_0)^2 \Gamma(2h + m)} {}_3F_2(2h - h_0, -m, -m; -h_0 - m + 1, -h_0 - m + 1; 1) \times \text{c.c.} \tag{36}$$

where $\lambda_{\phi;O} = g_O f_{\phi\phi O}$, $h = \Delta_\phi/2$, $h_0 = \Delta_O/2$.

The correction comes from the lowest symmetry singlet, *i.e.*, D. We define a cost function

$$\text{cost} = \sum_{\substack{0 \le \bar{m} \le m \le 4 \\ m + \bar{m} \le 5}} \left[ E^{(\text{num})}(\partial^m \bar{\partial}^{\bar{m}} o)/E_0 - \delta E(\partial^m \bar{\partial}^{\bar{m}} o; \lambda_{o;D}, \Delta_o^{(\text{ref})}, 3) - (\Delta_o + m + \bar{m}) \right]^2$$
$$+ \sum_{\substack{0 \le \bar{m} \le m \le 4 \\ m + \bar{m} \le 4}} \left[ E^{(\text{num})}(\partial^m \bar{\partial}^{\bar{m}} D)/E_0 - \delta E(\partial^m \bar{\partial}^{\bar{m}} D; \lambda_{D;D}, 3, 3) - (3 + m + \bar{m}) \right]^2 \tag{37}$$

and minimise with regard to $\Delta_o, E_0, \lambda_{o;D}, \lambda_{D;D}$, where $E^{(\text{num})}$ are the numerical excited energies measured by ED, the correction function $\Delta E$ is defined in Eq. (36).

# C  Complementary data

## C.1  DMRG convergence test

We test the energies $E(D)$ for different bound dimensions $D$ at different system sizes for different operators in different boundary conditions in the orbital space boundary scheme. We find good convergence at $D_{\max} = 6000$ for $N \le 64$, with the energy difference $E(D = 5000) - E(D_{\max} = 6000) \le 5 \times 10^{-5}$.

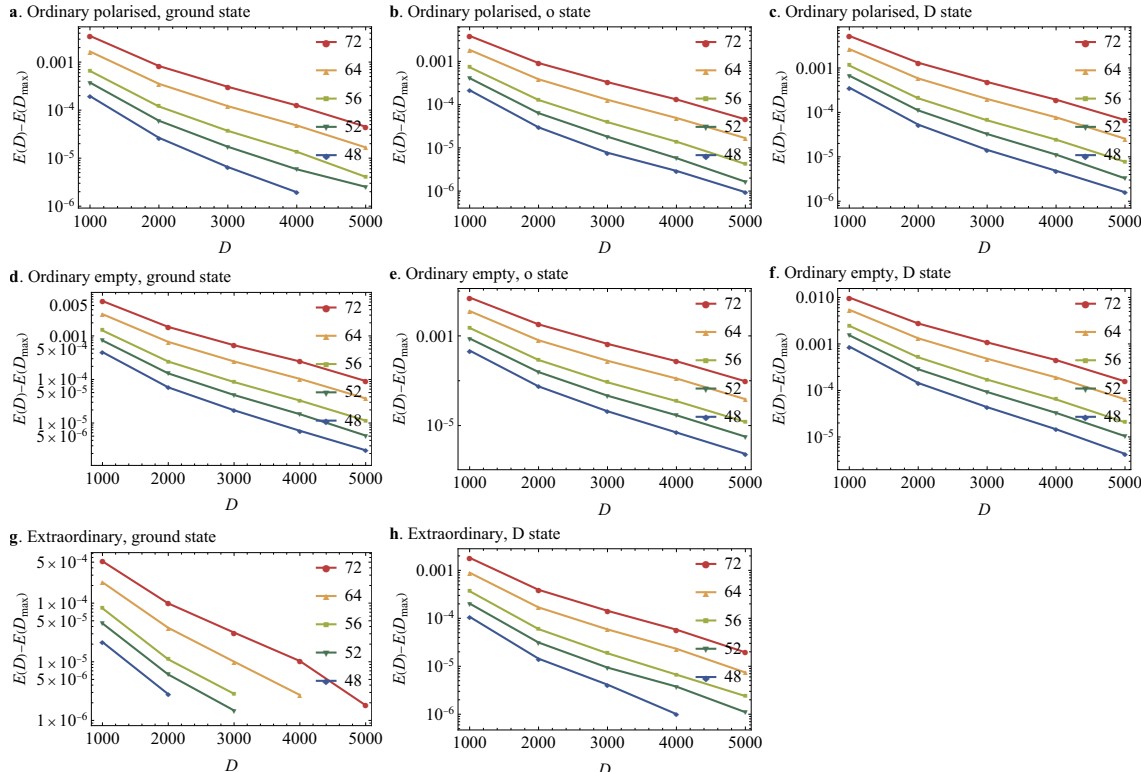

Figure 15: The energy difference $E(D) - E(D_{max})$ in the DMRG calculation between the energy at bond dimension $D$ and the maximal bond dimension $D_{max}$ in the calculation as a function of $D$ in the logarithmic scale at different system size for different operators in different boundary conditions in the orbital space boundary scheme.

## C.2 Scaling dimensions of the conformal multiplets

In this Appendix, we provide data of scaling dimensions of the conformal multiplets, in addition to Figures 3, 4 and 5

Table 3: The scaling dimensions of the conformal multiplets of $o$ and D in the orbital space boundary cut. The boundary condition is that the $m < 0$ orbitals are empty in the first three columns and $m < 0$ orbitals are half-filled and polarised to $x$-direction in the last column. The $N = 30$ raw and corrected data are plotted in Figure 4, and the $N \leq 64$ finite size scaling is plotted in Figure 3.

| Operator | $N = 30$ raw data | $N = 30$ corrected | $N \leq 64$, $m < 0$ empty estimation & scaling | $N \leq 64$, $m < 0$ polarised estimation & scaling |
|---|---|---|---|---|
| $o$ | 1.022 | 1.201 | $1.22(6) \ -1.09N^{-1/2}$ | $1.24(5) \ +0.89N^{-1/2}$ |
| $\partial o$ | 1.999 | 2.223 | $2.17(5) \ -0.89N^{-1/2}$ | $2.22(11) +1.79N^{-1/2}$ |
| $\partial\bar{\partial}o$ | 2.709 | 3.150 | $3.09(12) -2.05N^{-1/2}$ | $3.34(14) +2.30N^{-1/2}$ |
| $\partial^2 o$ | 3.049 | 3.243 | $3.18(4) \ -0.74N^{-1/2}$ | $3.6(1) \ -0.07N^{-1/2}$ |
| $\partial^2\bar{\partial}o$ | 3.808 | 4.305 | | |
| $\partial^3 o$ | 4.107 | 4.243 | | |
| $\partial^2\bar{\partial}^2 o$ | 4.500 | 5.191 | | |
| $\partial^3\bar{\partial}o$ | 4.883 | 5.395 | | |
| $\partial^4 o$ | 5.087 | 5.161 | | |
| $\partial^3\bar{\partial}^2 o$ | 5.477 | 6.257 | | |
| $\partial^4\bar{\partial}o$ | 5.826 | 6.350 | | |
| $\partial^3\bar{\partial}^3 o$ | 6.108 | 7.082 | | |
| D | 2.438 | 3.059 | $2.91(15) -2.57N^{-1/2}$ | $2.94(12) +2.01N^{-1/2}$ |
| $\partial$D | 3.435 | 4.142 | $3.89(15) -2.48N^{-1/2}$ | $4.02(14) +2.31N^{-1/2}$ |
| $\partial\bar{\partial}$D | 4.107 | 5.024 | | |
| $\partial^2$D | 4.012 | 4.838 | | |
| $\partial^2\bar{\partial}$D | 5.153 | 6.162 | | |
| $\partial^3$D | 5.072 | 5.906 | | |
| $\partial^2\bar{\partial}^2$D | 5.803 | 7.021 | | |
| $\partial^3\bar{\partial}$D | 5.835 | 6.967 | | |
| $\partial^4$D | 6.078 | 6.910 | | |

Table 4: The scaling dimensions of the conformal multiplets of D in normal surface CFT realised by the orbital space boundary condition that the $m < 0$ orbitals are polarised to $z$-direction. The $N = 30$ raw and corrected data are plotted in Figure 5b, and the $N \leq 64$ finite size scaling is plotted in Figure 5a. The error bar is estimated from finite-size extrapolation.

| Operator | $N = 30$ raw data | $N = 30$ corrected | $N \leq 64$ estimation & scaling |
|---|---|---|---|
| D | 2.151 | 2.982 | $2.91(25) - 4.30N^{-1/2} + 4.64N^{-3/2}$ |
| $\partial$D | 3.003 | 3.965 | $3.88(29) - 5.12N^{-1/2} + 9.52N^{-3/2}$ |
| $\partial\bar{\partial}$D | 3.759 | 4.959 | $4.77(27) - 4.75N^{-1/2} + 9.53N^{-3/2}$ |
| $\partial^2$D | 3.956 | 4.970 | $4.9\ (4)\ -6.45N^{-1/2} + 10.1N^{-3/2}$ |
| $\partial^2\bar{\partial}$D | 4.700 | 6.028 | |
| $\partial^3$D | 4.969 | 5.986 | |
| $\partial^2\bar{\partial}^2$D | 5.447 | 7.010 | |
| $\partial^3\bar{\partial}$D | 5.681 | 7.083 | |
| $\partial^4$D | 5.967 | 6.967 | |

## C.3 Extraction of bulk scaling dimensions from correlation functions

It is worth noting that from the bulk-to-surface two-point function $G_{\phi\hat{\phi}}$, the scaling dimension difference $\Delta_\phi - \Delta_{\hat{\phi}}$ can be extracted by calculating either its derivative at the north pole or its integral over the hemisphere.

$$\int_{\theta < \pi/2} d\Omega \, \frac{G_{\phi\hat{\phi}}(\Omega)}{G_{\phi\hat{\phi}}(+\hat{\mathbf{z}})} = \frac{1}{1 + \Delta_{\hat{\phi}} - \Delta_\phi}$$
$$\nabla^2 \log G_{\phi\hat{\phi}}(+\hat{\mathbf{z}}) = 2(\Delta_{\hat{\phi}} - \Delta_\phi). \tag{38}$$

The equations apply to the one-point function as well by setting $\hat{\phi}$ to the surface identity.

We show that all the correlators approach the CFT predictions with increasing system size. Quantitatively, we extract the scaling dimensions $\Delta_\epsilon$, $\Delta_o - \Delta_\sigma$ in ordinary surface CFT and $\Delta_\epsilon$,

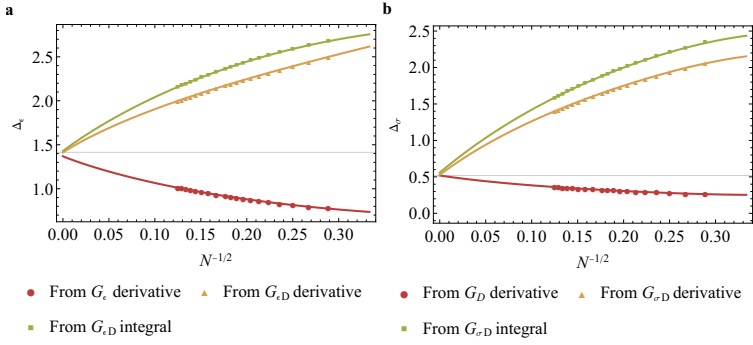

Figure 16: The finite-size results of the scaling dimensions (a) $\Delta_\epsilon$ and (b) $\Delta_\sigma$ extracted from the correlation functions calculated in the normal surface CFT in the orbital space boundary scheme, compared with the bootstrap result (grey gridline).

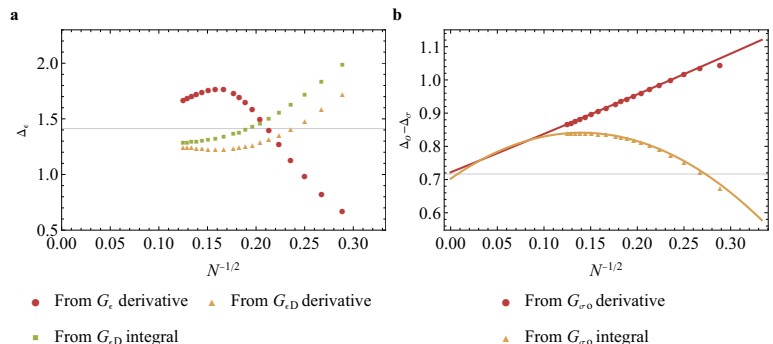

Figure 17: The finite-size results of the scaling dimensions (a) $\Delta_\epsilon$ and (b) $\Delta_o - \Delta_\sigma$ extracted from the correlation functions calculated in the ordinary surface CFT in the orbital space boundary condition that the $m < 0$ orbitals are half-filled and polarised in $x$-direction, compared with the bootstrap result (grey gridline).

$\Delta_\sigma$ in extraordinary surface CFT from the correlation functions. A finite-size scaling shows that the scaling dimensions agree with the values obtained by conformal bootstrap and state-operator correspondence in the thermodynamic limit.

## C.4 More correlation functions in the real-space boundary scheme

In this Appendix, we provide more correlation functions in the real-space boundary scheme, in addition to Figure 11.

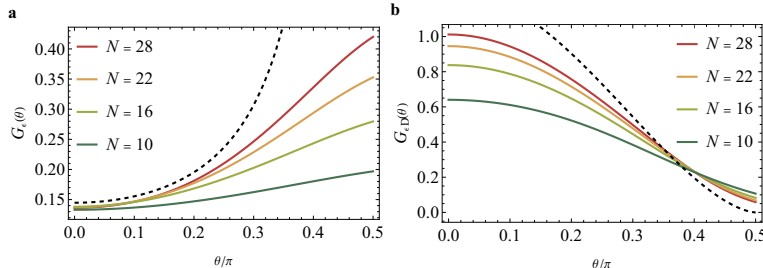

Figure 18: The one-point function (a) $G_\epsilon$ and the two-point functions (b) $G_{\epsilon D}$ in the ordinary surface CFT as a function of angle $\theta$ at finite size calculated in the real space boundary scheme, compared with the prediction by conformal symmetry (black dashed line). We have set $h_s = 10$.

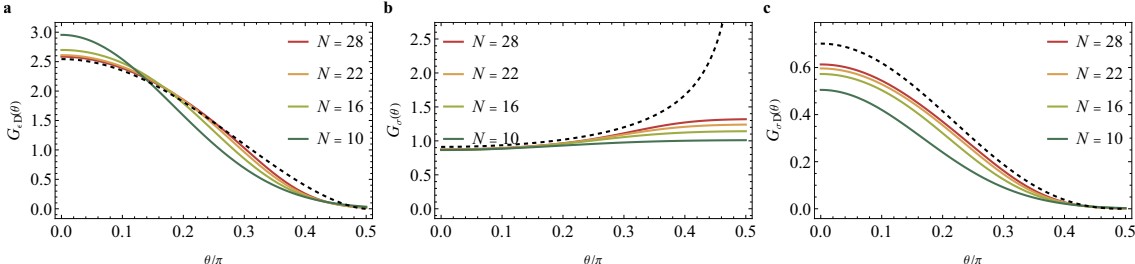

Figure 19: The one-point and two-point functions (a) $G_{\epsilon D}$, (b) $G_\sigma$ and (c) $G_{\sigma D}$ in the normal surface CFT as a function of angle $\theta$ at finite size calculated in the real space boundary scheme, compared with the prediction by conformal symmetry (black dashed line). We have set $h_s = 500$.

### C.5  Finite size correspondence between energy and entanglement spectrum

An interesting observation is the similarity between the energy spectrum of the ordinary surface CFT and the orbital space entanglement spectrum of the Ising CFT at finite size. In order to calculate the entanglement spectrum, we take the ground state at the critical point of the bulk Ising model on the fuzzy sphere. We divide it into two subsystems with all the $m > 0$ and $m < 0$ orbitals respectively, denoted $<$ and $>$, and calculate the reduced density matrix

$$\rho_{<>} = \text{tr}_< |0\rangle\langle 0| \tag{39}$$

The entanglement spectrum is thus the eigenvalues of $-\log \rho_{<>}$. Like the energy spectrum, each level in the entanglement spectrum carries quantum numbers $\mathbb{Z}_2$, $n_{e,>}$ and $l_{z,>}$. We pick out sector with half-filling $n_{e,>} = N/2$. In this sector, the particle-hole symmetry sends a state with $l_{z,>}$ to a state with $N^2 - l_{z,>}$. We thus shift the quantum number $l_{z,>}$ by $N^2/2$ so that the spectrum is symmetric reflecting with respect to $l_{z,>}$ axis like in the energy spectrum. We observe that the entanglement spectrum forms a certain uniform spacing structure, *e.g.*, the spacings between lowest levels for each $l_z$ are almost the same. It is natural to ask if it corresponds to the energy spectrum of some quantum system described by CFT. The most natural candidate is the ordinary surface CFT that could be realised by removing half of the orbitals and keeping the Hamiltonian the same as the bulk. In order to compare the two spectra, we move the lowest levels to 0 and rescale the second lowest level to 1. The result is shown in Figure 20. The energy spectrum levels (symbols) and the entanglement spectrum levels (bar) are almost in one-to-one correspondence and have very close values up to $\Delta/\Delta_1 = 4$.

However, we also note that this correspondence may not survive the finite-size scaling. As we increase the system size, some of the levels on the entanglement spectrum start to deviate from the corresponding value of the ordinary surface CFT operators. Hence, this correspondence may be just a coincidence at a certain finite size.

ref.bbl

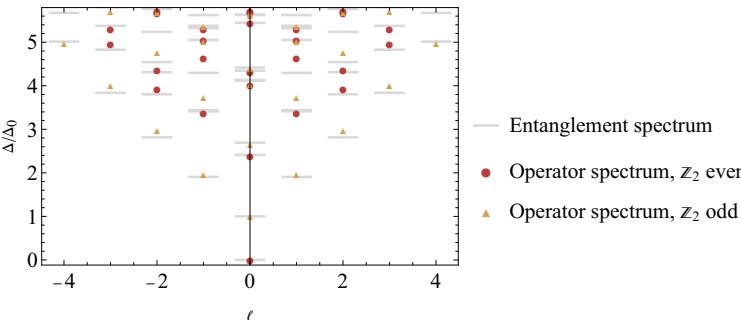

Figure 20: The comparison between the orbital space entanglement spectrum at $N = 16$ and the energy spectrum at $N = 30$. We rescale the lowest level to 0 and the second lowest level to 1.

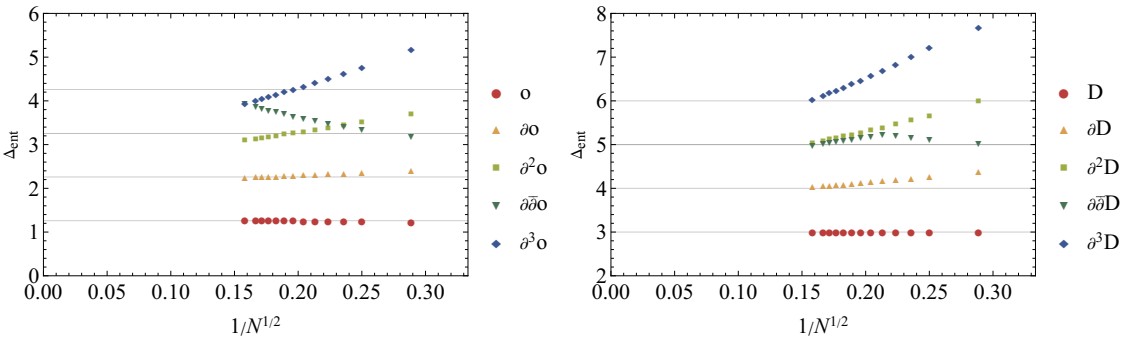

Figure 21: The finite size scaling of the lowest (left) $\mathbb{Z}_2$-odd and (right) $\mathbb{Z}_2$-even levels on the entanglement spectrum. The grey lines indicate the corresponding scaling dimensions of the ordinary surface CFT operators, calibrated such that for the second-lowest $\mathbb{Z}_2$-even state that supposedly corresponds to D, $\Delta_{\text{ent}} = 3$.

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
