# Peer review of "Studying the 3d Ising surface CFTs on the fuzzy sphere"

_SciPost Physics_

## Round 3 · Referee Report · Anonymous (Referee 1) · 2024-9-20

Strengths

  1. The paper introduces a new, exciting application for fuzzy sphere, a recently developed powerful tool in understanding CFTs. The power of their technique is clearly illustrated and the paper has good alignment with previous results.

  2. The paper provides multiple checks on their technique for studying BCFTs and gives the reader confidence in the validity of their numerical calculations.

  3. Their technique is easily applicable to other BCFTs provided the bulk CFT can be placed on the fuzzy sphere practically.

Weaknesses

  1. The introduction could be improved from its current state. Some quantities are not explicitly defined, and thus the introduction does not flow as well. Additionally, the penultimate paragraph could better emphasize the results of the paper. This paragraph does not state the main takeaway of the paper as directly as e.g. the second sentence of the abstract does. Additionally, this paragraph does not motivate well why two schemes are used and the takeaway from using both schemes.

  2. The lack of explicit definitions continues throughout the paper and makes some sections a little hard to follow. Additionally, some sections could do a better job of sign-posting.

  3. The grammar is not perfect (some examples are listed below, although the list is not comprehensive). This weakness is not major, but it could be improved for the final version of the paper.

Report

This paper presents a method for using the fuzzy sphere to study BCFTs, specifically the 2+1D Ising CFT with a boundary at the ordinary and normal fixed points. The paper computes scaling dimensions and OPE coefficients for boundary operators for both fixed points. Additionally, the paper provides nonperturbative calculations of the boundary central charge at both fixed points.

The paper's technique is an important development for the field of BCFT. Their technique seems applicable to other CFTs that can be studied with the fuzzy sphere.

However, the lack of definitions and clarity in some parts of the paper hinder its effectiveness in conveying its message. Upon improving clarity, this paper should be well-suited for this journal.

Requested changes

  1. General grammar/spelling comments (this list is not comprehensive, but should be helpful in guiding edits):

There is at least one instance where plural subjects are accompanied by singular verbs and vice versa -- namely "there exist a set..." in the second paragraph.

In the second paragraph of the paper, "the CFT" should be "a CFT".

In the paragraph above Eq. (3), the sentence "First, acting ..." is clunky.

The paper sometimes uses the future tense where the present tense is more appropriate.

  1. In the second paragraph of the paper, a few things are not defined. For example, the paper never explicitly states that we are considering 2+1D CFTs. The paper also would benefit from briefly defining scaling dimensions, spin, primaries, and descendants -- the standard for papers in this field of this length seems to be to define these terms and give a brief few sentence introduction to CFTs.

  2. The data in the paragraph at the start of page 4 could be placed in a table. This change would make it easier to compare previous results with their results. Additionally, they should state that the large-N calculations are for the O(N) model. Finally, the paper should consider briefly defining the $\sigma$, $\epsilon$, $\epsilon'$, and $D$ operators -- or at least stating for example that $D$ is the displacement operator -- as it seems that defining them is the convention in the field for papers of this length.

  3. As mentioned in the previous section, directly stating the paper's significance would improve the last paragraph of page 4. Additionally, the paper should comment on why it chose to calculate quantities using two schemes and which scheme should be preferred.

  4. On page 7, the paper should briefly explain why a +x direction magnetic field corresponds to a free boundary condition.

  5. On page 8, is the O(2) symmetry the same as the U(1) symmetry?

  6. On page 9 right before the start of section 3.1, the paper should explicitly state which method it uses (finite size scaling or Lao and Rychkov's method presented in App. B). Additionally, App. B likely should also cite Lao and Rychkov's paper and should signpost and refer back to section 3 to help the reader.

  7. On page 12, the paper should state explicitly how it uses the density operator to compute one and two-point functions -- especially because using fuzzy sphere to calculate OPE coefficients is a relatively new calculation that not many readers will be familiar with. The paper should also mention that it uses finite size scaling to determine and subtract off the $\lambda$ coefficients in Eq. (17). Additionally, the paper does not clearly state if Eq. (14) is what it means by "CFT predictions" in Sec 4.1 (at least, that is what I assumed after reading the paper). If Eq. (14) is indeed what the paper means by CFT predictions, what OPE coefficient value does the paper use for the black dashed line in the plots in Fig. 6?

  8. On page 17, the paper could explain the relationship between <pol|I> and Z a little more explicitly.

  9. Appendix A should also signpost -- i.e., explain its purpose and say where Eq. (25) comes from for ease of reading. Appendix C.3 could also do this (e.g., it could give an equation reference for $G_{\phi\hat \phi}$ so the reader can more easily confirm Eq. (38).

Recommendation

Ask for minor revision

  • validity: top
  • significance: high
  • originality: high
  • clarity: ok
  • formatting: excellent
  • grammar: good

Author:  Zheng Zhou  on 2024-11-07  [id 4946]

(in reply to Report 1 on 2024-09-20)

(1) General grammar/spelling comments (this list is not comprehensive, but should be helpful in guiding edits): - There is at least one instance where plural subjects are accompanied by singular verbs and vice versa -- namely 'there exist a set\dots' in the second paragraph. - In the second paragraph of the paper, 'the CFT' should be 'a CFT'. - In the paragraph above Eq. (3), the sentence 'First, acting ...' is clunky. - The paper sometimes uses the future tense where the present tense is more appropriate.

Thank you for your recommendation and for your valuable comments. We believe that the changes made have further improved our paper. We have fixed the issues listed in the revised manuscript and done a careful proofreading.

(2) In the second paragraph of the paper, a few things are not defined. For example, the paper never explicitly states that we are considering (2+1)D CFTs. The paper also would benefit from briefly defining scaling dimensions, spin, primaries, and descendants --- the standard for papers in this field of this length seems to be to define these terms and give a brief few sentence introduction to CFTs.

Thank you for the suggestions. We have added some descriptions accordingly to the introduction section.

(3) The data in the paragraph at the start of page 4 could be placed in a table. This change would make it easier to compare previous results with their results. Additionally, they should state that the large-$N$ calculations are for the $\mathrm{O}(N)$ model. Finally, the paper should consider briefly defining the $\sigma$, $\epsilon$, $\epsilon'$, and $D$ operators --- or at least stating for example that $D$ is the displacement operator --- as it seems that defining them is the convention in the field for papers of this length.

Thank you for the suggestions. We have organised the previous results into Table 1 as a comparison to our results. We have also added the statement about large-$N$ calculation and description for the mentioned operators.

(4) As mentioned in the previous section, directly stating the paper's significance would improve the last paragraph of page 4. Additionally, the paper should comment on why it chose to calculate quantities using two schemes and which scheme should be preferred.

Thank you for the suggestions. We have added several sentences to the paragraph to emphasise these points: 'As we will show, the conformal data extracted from the real and orbital space boundary schemes are consistent, demonstrating that they realise the same boundary CFT.', 'Our main results are listed in Table 1. These results are calculated from the orbital space boundary scheme, where much larger system sizes can be accessed than the real space boundary scheme. Some of these results are reported for the first time with non-perturbative methods, such as the OPE coefficient $b_{\epsilon\mathrm{D}}$ in the ordinary surface CFT, and the boundary central charge $c_\mathrm{bd}$ in ordinary and normal surface CFTs.'

(5) On page 7, the paper should briefly explain why a $+x$ direction magnetic field corresponds to a free boundary condition.

A more precise statement is that when a magnetic field in $x$-direction is imposed, the $\mathbb{Z}_2$-symmetry on the boundary is preserved and the boundary is disordered. Its flow to ordinary surface CFT is observed numerically. We have revised the wording in the paragraph.

(6) On page 8, is the $\mathrm{O}(2)$ symmetry the same as the $\mathrm{U}(1)$ symmetry?

The $\mathrm{U}(1)$ symmetry we mentioned is identical to the $\mathrm{SO}(2)$ rotation along the $z$-axis, and the parity symmetry supplies another $\mathbb{Z}_2$, enlarging the rotation symmetry from $\mathrm{SO}(2)$ to $\mathrm{O}(2)$. In the revised manuscript, we have modified $\mathrm{U}(1)$ to $\mathrm{SO}(2)$ for consistency.

(7) On page 9 right before the start of section 3.1, the paper should explicitly state which method it uses (finite size scaling or Lao and Rychkov's method presented in App. B). Additionally, App. B likely should also cite Lao and Rychkov's paper and should signpost and refer back to section 3 to help the reader.

Thank you for the suggestions. We have made the revisions accordingly.

(8) On page 12, the paper should state explicitly how it uses the density operator to compute one and two-point functions -- especially because using fuzzy sphere to calculate OPE coefficients is a relatively new calculation that not many readers will be familiar with. The paper should also mention that it uses finite size scaling to determine and subtract off the $\lambda$ coefficients in Eq. (17). Additionally, the paper does not clearly state if Eq. (14) is what it means by 'CFT predictions' in Sec 4.1 (at least, that is what I assumed after reading the paper). If Eq. (14) is indeed what the paper means by CFT predictions, what OPE coefficient value does the paper use for the black dashed line in the plots in Fig. 6?

Thank you for the suggestions. We have added descriptions for using the density operators to calculate the bulk-to-surface OPE coefficients and do finite-size scaling. When applying Eq. (14), the bulk-to-surface OPE coefficients $a_\phi$ and $b_{\phi\hat{\phi}}$ are taken to be values extrapolated on fuzzy sphere (Table 2). We have also added a footnote to the manuscript.

(9) On page 17, the paper could explain the relationship between $\langle\textrm{pol}|\mathbb{I}\rangle$ and $Z$ a little more explicitly.}

We have added a more detailed explanation in the revised manuscript. We consider the overlap between some state $|\Phi_0(\mathbf{r})\rangle$ in the Hilbert space of the sphere and the ground state of the CFT. It can be represented by the path integral [Zou, Phys. Rev. B 105, 165420 (2022)]

$$ \langle\Phi_0(\Omega)|\mathbb{I}\rangle\sim\int_{\tau<0,S^2\times\mathbb{R}_{-},\Phi(\tau=0,\Omega)=\Phi_0(\Omega)}\mathscr{D}\Phi\,e^{-S[\Phi]}. $$

Here $\mathscr{D}\Phi$ is the functional measure, and $\sim$ means that a normalisation factor that does not depend on the radius is omitted. This represents a CFT living on the $\tau<0$ spacetime. The choice of $|\Phi_0(\mathbf{r})\rangle$ determines the boundary condition at $\tau=0$. Through a Weyl transformation, the interface on the cylinder is transformed into the unit sphere on the flat spacetime with the CFT living in the interior, and the functional integral will give the partition function of a surface CFT living on a ball in flat spacetime with radius 1.

$$ \langle\Phi_0(\mathbf{r})|\mathbb{I}\rangle\sim\int_{r<1,\mathrm{flat},\Phi(\hat{\mathbf{n}})=\Phi_0(\hat{\mathbf{n}})}\mathscr{D}\Phi\,e^{-S[\Phi]}\sim Z(\textrm{surface CFT on }S^2). $$

The boundary condition of the surface CFT is given by $|\Phi_0(\Omega)\rangle$. The scaling behaviour of this partition function is related to the boundary central charge.

(10) Appendix A should also signpost -- i.e., explain its purpose and say where Eq. (25) comes from for ease of reading. Appendix C.3 could also do this (e.g., it could give an equation reference for $G_{\phi\hat{\phi}}$ so the reader can more easily confirm Eq. (38).)

Thank you for the suggestions. We have made the revisions accordingly.

---

## Round 3 · Referee Report · Anonymous (Referee 2) · 2024-9-23

Strengths

1- Important and timely subject 2- Novel and creative method, new results

Weaknesses

1- Results could have been more thorough 2- Some issues with methodology (see report)

Report

This is a very nice application of the fuzzy sphere regulator of the Ising model to boundary CFT in d=3, and the results should hopefully encourage many future studies further developing the method. Many previously known results are reproduced as a test of the method, and some new results are obtained demonstrating that is has concrete advantages.

Some comments:

1- It is stated below equation (10) that the value of the speed of light is identical in the bulk CFT and the boundary CFT, and so can be calibrated by setting the stress tensor dimension to be 3. I am not exactly sure how the authors implemented this in practice since the bulk stress tensor as a state arises for the CFT on a sphere without boundary, whereas they are calculating the spectrum in the CFT with a boundary, but I am guessing that what they did was to calculate the spectrum on the sphere using the same microscopic UV parameters and assume that the rescaling factor between energy and dimension is the same for the case with a boundary. However, the presence of the boundary will modify the Hamiltonian, and therefore modify the rescaling factor. So it seems to me that they really should be directly rescaling the energies on the theory with boundary to fix the dimension of the displacement operator to be 3, and this would improve the accuracy of their results. If the authors disagree with this approach then perhaps they could explain why.

2- The method by which the authors estimate their errors is to take the difference between the extrapolated value (at infinite radius) and the last computed value. This seems like a significant overestimate of the error, and therefore it seems at least a little concerning that for example the known dimension of the operator o is at the edge of their error bars. Can the authors also estimate the error by performing the fit in different ways (perhaps by using different subsets of the finite N results) and seeing if this is consistent with the errors they choose? Moreover, it would be helpful for the authors to explain the rationale behind the powers of N in the fits in table 2 and Fig 14.

3- In equation (12), they include the descendant operator $\partial \bar{\partial}D$, but the contribution to the energies of states from total derivatives should vanish. On the other hand, as they point out, there is an operator of dimension 5.02, which is so close to 5 that it will behave nearly the same ($\sim N^{-3/2}$) as the term they write down.

4 - At the bottom of page 13 and top of page 14, they sometimes use $\Delta$ and sometimes use $\Delta_\phi$ for the dimension of $\phi$, but it would be better to be consistent.

5- Could the authors explain why setting the m<0 orbitals to be empty also produces the ordinary surface CFT (why 1. and 2. below equation 9 are equivalent)?

6- The authors do not do the special transition, but merely comment in the discussion on how it should be possible. Could the authors comment on why they chose not to do the special transition, and how difficult in their opinion it would have been to do (i.e. some discussion of the challenges involved if there was some reason it would have been hard). Also I think it would be good to have this discussion in the introduction in order to make it clear up front that the authors did not study the special transition.

7- At the bottom of page 28 there is a stray "ref.bbl"

Recommendation

Publish (surpasses expectations and criteria for this Journal; among top 10%)

  • validity: high
  • significance: high
  • originality: high
  • clarity: good
  • formatting: excellent
  • grammar: perfect

Author:  Zheng Zhou  on 2024-11-07  [id 4947]

(in reply to Report 2 on 2024-09-23)

This is a very nice application of the fuzzy sphere regulator of the Ising model to boundary CFT in $d=3$, and the results should hopefully encourage many future studies further developing the method. Many previously known results are reproduced as a test of the method, and some new results are obtained demonstrating that is has concrete advantages.

Thank you for your recommendation and for your valuable comments. We believe that the changes made have further improved our paper.

Some comments: (1) It is stated below equation (10) that the value of the speed of light is identical in the bulk CFT and the boundary CFT, and so can be calibrated by setting the stress tensor dimension to be 3. I am not exactly sure how the authors implemented this in practice since the bulk stress tensor as a state arises for the CFT on a sphere without boundary, whereas they are calculating the spectrum in the CFT with a boundary, but I am guessing that what they did was to calculate the spectrum on the sphere using the same microscopic UV parameters and assume that the rescaling factor between energy and dimension is the same for the case with a boundary. However, the presence of the boundary will modify the Hamiltonian, and therefore modify the rescaling factor. So it seems to me that they really should be directly rescaling the energies on the theory with boundary to fix the dimension of the displacement operator to be 3, and this would improve the accuracy of their results. If the authors disagree with this approach then perhaps they could explain why.

The speed of light is the same in the bulk and on the boundary. Adding a boundary or defect term to the Hamiltonian does not change the speed of light, as long as Lorentz invariance is emerged on the boundary or defect. The identification of the speed of light in bulk and line defect has been discussed in a previous paper [Hu et al., Nat. Commun. 15, 3659 (2024)]. The same conclusion applies to our case of boundaries. A theoretical argument of this statement is that the bulk CFT operators are realised on fuzzy sphere by the same local operators with or without boundaries. Subsequently, the expression for the component $T^{00}$ of the bulk stress tensor, which integrates to be the generator of dilatation, is the same in the bulk CFT as in the surface CFT.

In practice, we first simulate the bulk model on the fuzzy sphere and extract the energy of bulk ground state and bulk stress tensor to calculate the speed of light and then move to the model with boundary at the same parameter choice of $U_0,U_1$ and $h$. We have also added the results for the equivalence of different calibrations in Appendix C.2. Here we choose to use the bulk calibration because not assuming a displacement operator with $\Delta_\mathrm{D}=3$ is more objective, and its existence can be used as an extra evidence of conformal symmetry.

(2) The method by which the authors estimate their errors is to take the difference between the extrapolated value (at infinite radius) and the last computed value. This seems like a significant overestimate of the error, and therefore it seems at least a little concerning that for example the known dimension of the operator $o$ is at the edge of their error bars. Can the authors also estimate the error by performing the fit in different ways (perhaps by using different subsets of the finite $N$ results) and seeing if this is consistent with the errors they choose? Moreover, it would be helpful for the authors to explain the rationale behind the powers of $N$ in the fits in Table 2 and Fig. 14.

We have added various different estimations for the scaling dimension $\Delta_o$ in Appendix C.2. Most of these extrapolated results fall between $1.22$ and $1.28$ towards the higher end and closer to the known results by Monte Carlo and conformal bootstrap. We hope that with these additional results, you can feel more reassured about our results. The deviation between our estimation and the known results may come from our negligence of higher order corrections. We choose not to include these terms to avoid overfitting.

We have also added explanations for the correction powers in Table 2 and Figure 14. For the bulk-to-boundary OPE coefficients in Table 2, the subleading contributions for bulk $\epsilon$ come from its descendants $\partial^\mu\epsilon$ and stress tensor $T^{\mu\nu}$ with a power of $-1/2$ and $-(3-\Delta_\epsilon)/2=-0.79$; the subleading contributions for bulk $\sigma$ comes from its descendants $\partial^\mu\sigma$ and $\partial^\mu\partial^\nu\sigma$ with a power of $-1/2$ and $-1$. For Figure 14, the leading finite size effect comes from the irrelevant boundary operator $\mathrm{D}$ with power law corrections $N^{-(\Delta_\mathrm{D}-2)/2}=N^{-1/2}$ and the irrelevant bulk operator $C^{\mu\nu\rho\sigma}$ with $N^{-(\Delta_{C^{\mu\nu\rho\sigma}}-3)/2}=N^{-1.01}$. We have also revised our results.

(3) In equation (12), they include the descendant operator $\bar{\partial}\partial D$, but the contribution to the energies of states from total derivatives should vanish. On the other hand, as they point out, there is an operator of dimension 5.02, which is so close to 5 that it will behave nearly the same ($\sim N^{-3/2}$) as the term they write down.}

Thank you for pointing out that $\bar{\partial}\partial D$ does not contribute. We have thought more carefully about that. The finite-size corrections comes by irrelevant surface primary $\hat{S}$ in the symmetry singlet sector and irrelevant $\mathbb{Z}_2$-even spin-even bulk primaries $S$

$$ \Delta_{\hat{\phi}}(N)=\Delta_{\hat{\phi}}+\sum_{\hat{S}}\lambda_{\hat{S};\hat{\phi}}R^{2-\Delta_{\hat{S}}}+\sum_S\lambda_{S;\hat{\phi}}R^{3-\Delta_S}. $$

For the Ising surface CFTs, the lowest correction comes from the displacement operator $\mathrm{D}$ with scaling dimension $\Delta_\mathrm{D}=3$. The lowest irrelevant bulk corrections from $\epsilon$ is tuned away, and the second lowest is $C^{\mu\nu\rho\sigma}$ with scaling dimension $\Delta_{C^{\mu\nu\rho\sigma}}=5.02$. Thus,

$$ \Delta_{\hat{\phi}}(N)=\Delta_{\hat{\phi}}+\lambda_{\mathrm{D};\hat{\phi}}N^{-1/2}+\lambda_{C^{\mu\nu\rho\sigma};\hat{\phi}}N^{-(\Delta_{C^{\mu\nu\rho\sigma}}-3)/2}+\mathcal{O}(N^{-(\Delta_{C^{\mu\nu\rho\sigma}}-3)/2}) $$

So the lowest terms should be $N^{-1/2}$ and $N^{-1.01}$. We have also revised our results.

(4) At the bottom of page 13 and top of page 14, they sometimes use $\Delta$ and sometimes use $\Delta_\phi$ for the dimension of $\phi$, but it would be better to be consistent.}

Thank you for pointing that out. We have modified the notation and use $\Delta_\phi$ in the revised manuscript.

(5) Could the authors explain why setting the $m<0$ orbitals to be empty also produces the ordinary surface CFT (why 1. and 2. below equation 9 are equivalent)?

In both cases of $m<0$ orbitals empty and polarised to $+x$, the $\mathbb{Z}_2$-symmetry on the boundary is preserved and the boundary is disordered. The ordinary surface CFT is stable and requires no fine-tuning of boundary parameters. Thus, we expect that the difference between those two choices is just the tuning of an irrelevant boundary operator. Also, their flows to ordinary surface CFT are observed numerically. We have also revised the wording of the paragraph.

(6) The authors do not do the special transition, but merely comment in the discussion on how it should be possible. Could the authors comment on why they chose not to do the special transition, and how difficult in their opinion it would have been to do (\textit{i.e.} some discussion of the challenges involved if there was some reason it would have been hard). Also I think it would be good to have this discussion in the introduction in order to make it clear up front that the authors did not study the special transition.

The special surface CFT can be realised in a similar scheme by applying boundary conditions in orbital space, although the detailed numerical study is beyond the scope of this work. In the orbital space boundary scheme, the pinning of the $m<0$ orbitals is equivalent to imposing a tunable boundary polarisation term. More specifically, removing the pinned orbitals can be realised by adding a magnetic field on the remaining orbitals that is proportional to the polarisation $n_{m^<}^\alpha=\langle\mathbf{c}^\dagger_{m^<}\sigma^\alpha\mathbf{c}_{m^<}\rangle$ of the removed orbital $m^<$, where $\alpha=0,x,y,z$. The polarisation vector $n_{m^<}^\alpha$ serves as tuning parameters in the surface CFT. In the paper, we have shown that both $n_{m^<}^x=0$ and $n_{m^<}^x=1$ correspond to the ordinary surface CFT. The choice $n_{m^<}^x=-1$ leads to a spontaneous symmetry breaking on the surface and is likely to correspond to the extraordinary surface CFT. By fine-tuning $-1<n_{m^<}^x<0$, one would find a surface critical point that is likely to correspond to the special surface CFT. Yet, the special surface CFT is an unstable fixed point, so one has to tune the parameter accurately.

The detailed study of surface CFT is left for our future work because of the necessity of extra fine-tuning. However, we have discussed the method in the Discussion section. In the revised manuscript, we have also added several sentences in the abstract, introduction and discussion to emphasize and clarify this point.

(7) At the bottom of page 28 there is a stray 'ref.bbl'

Thank you for pointing that out. We have fixed the mistake in the revised manuscript.

---

## Editorial Decision

resubmitted